# High-Precision Measurement of Height Differences from Shadows in Non-Stereo Imagery: New Methodology and Accuracy Assessment

Camilo Andrés Rada Giacaman [1,2]

1   Centro de Investigación GAIA Antártica (CIGA), Universidad de Magallanes,
    Punta Arenas 6210427, Chile; camilo.rada@umag.cl
2   Department of Earth, Ocean and Atmospheric Sciences, University of British Columbia,
    Vancouver, BC V6T 1Z4, Canada

**Abstract:** The shadow-height method has been extensively used to extract the heights of buildings from the shadows they cast in non-stereo (single view) aerial and satellite imagery. However, the use of this method in Earth sciences has been limited, partially due to the relatively low accuracy reported, the fuzziness of shadow edges, the complexities of the scanning sensors, and a lack of software tools. In this paper, we present an enhanced shadow-height methodology offering significant accuracy improvement. These improvements are mainly the result of using a physical approach to model the illumination gradient through the edge of shadows and by leveraging meteorological data to precisely estimate atmospheric refraction. We validated 91 shadow-derived height estimations from images obtained by the Advanced Spaceborne Thermal Emission and Reflection Radiometer (ASTER) at three sites with latitudes between 33 and 78°S: The Andes Mountains, Sentinel Range, and Abbot ice shelf. Reference measurements were obtained from Global Navigation Satellite System (GNSS) surveys and the Ice, Cloud, and land Elevation Satellite (ICESat). The observed errors fell below 6% for small height differences ($\sim$20 m) and below 2% for larger height differences ($\gtrsim$300 m). Our validation data cover solar elevations ranging from 3.7 to 42.2°, and we observed smaller absolute errors at lower solar elevations. This novel information can be valuable for studying surface elevation changes in present and old imagery and extending glacier volume variation time-series.

**Keywords:** shadows; shadow-height; freeboard height; ice shelves; elevation changes; glacier variations; volcanic plumes; ASTER





## 1. Introduction

Research on Earth sciences often requires the measurement of terrain elevation changes. One important example of this kind of application is glacial mass balance, where glacier area and thickness changes are required to quantify ice mass variations. For these studies and a wide range of other applications, surface topography has been mapped intensively using Radar altimetry, Interferometric Synthetic Aperture Radar (InSAR), Light Detection and Ranging (LiDAR), photogrammetry, and Global Navigation Satellite System (GNSS) surveys, among other techniques. However, topographic data from these sources become progressively scarce for dates further back in time. Such data availability limitations could be mitigated by using the shadow-height method on non-stereo (single look) satellite or aerial images. Non–stereo imagery is arguably the most common and readily available type of remotely sensed data, with the highest spatial resolution and more extensive temporal coverage. Space-borne imagery records extend to the 1970s for civilian satellites (i.e., Landsat) or the 1950s for (now declassified) military satellites (e.g., Corona).

The shadow-height method takes advantage of the basic geometrical properties of shadows often present in satellite and aerial imagery. Figure 1 shows the simplest conceptual form of this method, where **L** represents the horizontal length of the shadow and θ

is the solar elevation angle. In this configuration, the height of the projector $\Delta h$ is given by $\Delta h = \mathbf{L} \tan \theta$. Unfortunately, this simple approach cannot be applied in most remote sensing settings, where the perspective of the image generally does not allow for the direct measurement of the horizontal shadow length $\mathbf{L}$. What can be directly measured from a remote sensing platform is the angular size of the shadow $\phi$, as shown in Figure 2. Suppose we know the positions of the Sun ($\odot$), the satellite ($O$), the projector ($P$), and the angular size $\phi$ of the shadow. Then, we can calculate the corresponding position of the shadow ($S$) as the point on the line $\odot$–$P$ at an angular distance $\phi$ from $P$, as seen from the satellite. Knowing the three-dimensional position of $S$ relative to $P$, the height difference $\Delta h$ can be derived. This calculation relies on elementary geometric concepts, but its application has many technical obstacles. In this paper, we present a high-accuracy methodology to perform this calculation, allowing for the extraction of new and valuable data from available archive imagery.

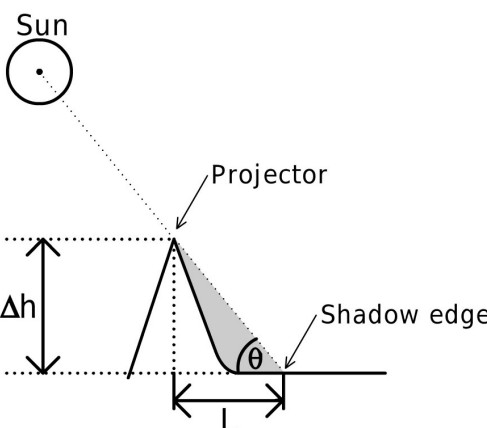

**Figure 1.** Geometric schematic of a simple height difference calculation using a shadow.

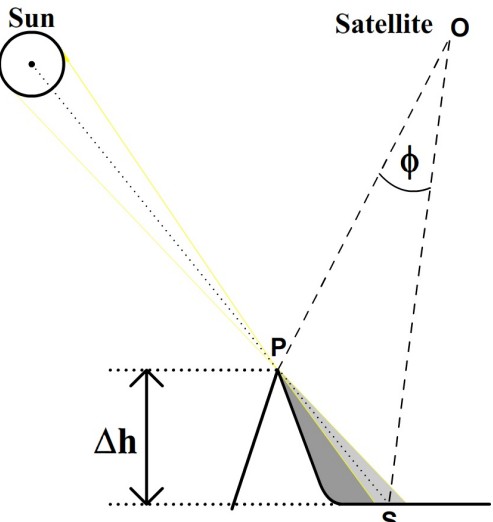

**Figure 2.** Schematic geometry of height difference calculations using shadows. The shadow $S$ is cast by the projector $P$, and it is known to lie on the line $\odot$–$P$. We can calculate the 3D position of the shadow ($S$) by using the angular size of the shadow from the satellite $\phi$.

For several decades, the shadow-height method has been part of the remote sensing toolbox. However, its use has remained mostly confined to the detection and characterization of buildings, initially on aerial images [1] and later being extended to spaceborne platforms [2–8] (among others). Similarly, the method has been applied to other man-made structures, such as pylons [9], as well as trees [10,11].

To date, applications of the shadow-height method for earth sciences have been scarce [3,12]. Nevertheless, it has been adapted and used for the measurement of ice shelf freeboard heights [13,14] and the height of volcanic plumes [12]. The present work provides a methodological improvement over previous implementations by presenting a careful treatment of atmospheric refraction and a physical approach to shadow modeling and fitting. We also perform a comprehensive error assessment and field validation. Note that the work presented here is based on the Master's thesis of the author [15], dated 2009.

Two factors that might have affected the limited use of the shadow-height method in the earth sciences literature are the large associated errors reported and its reduced applicability at tropical latitudes, where solar elevations are always too high to produce significant shadows. We present multiple methodological improvements that increase the accuracy of shadow-derived height measurements. Regarding the scope of the method, we emphasize that, at mid to high latitudes, shadows are common and often present in satellite imagery of mountainous terrain. This affirmation is true even for sun-synchronous satellite platforms—an orbit that is, in most cases, optimized to minimize the presence of such shadows. Our first methodological contribution is related to the treatment of atmospheric refraction and leveraging re-analysis atmospheric data to constrain its magnitude. Atmospheric refraction has often been neglected in shadow-height calculations; however, this effect can have a significant impact at very low solar elevations, such as those used to measure ice shelf or iceberg freeboard heights. Our second most significant methodological contribution is a physical approach to identify the exact locations of shadows in images. This approach takes advantage of the diffuse nature of shadow edges to achieve fine sub-pixel positioning accuracy.

At high- and mid-latitudes during the winter, snow accumulation and glacier thickness variations are the most common sources of terrain change. These factors evolve at time scales much shorter than changes observed in the elevation of the surrounding mountains, summits, and ridges. In particular, the elevation difference between a rocky summit and the surrounding bedrock topography is relatively stable at decadal to centennial scales. Erosion and sedimentation rates by water and glaciers are usually on the order of millimetres to centimetres per year [16], while glacier thickness changes are often between a few to several meters per year. For example, the O'Higgins glacier in Patagonia has been observed since 1914, for which thinning rates up to 6.7 m y$^{-1}$ have been shown [17]. These significant thinning rates set a relatively low accuracy requirement for useful elevation change measurements. Additionally, given the significant difference in the rate of change of mountains and glaciers, we can safely attribute any changes in their relative height to variations in the surface elevation of glaciers.

Topographic information recovered from old imagery using the shadow-height method could be instrumental in supplementing the scarce quantitative glacier elevation measurements available in the second half of the 20th century. Many other research fields could also benefit from the capacity of this improved shadow-height method to extract novel topographic data from archive imagery.

On aerial imagery or analog spaceborne platforms, such as the U.S. Corona or Hexagon spy satellites, we can directly use the geometry shown in Figure 2; however, this approach cannot be applied to digital satellite platforms with scanning sensors (e.g., whisk broom or push broom scanners) such as Landsat, SPOT, IKONOS, or the Advanced Spaceborne Thermal Emission and Reflection Radiometer (ASTER). This limitation arises from the movement of the satellite during the acquisition; therefore, the projector and the shadow are often imaged at different times and from different positions. In the case of a shadow projected in the along-track direction, the observation points of the projector ($P$) and shadow ($S$) are located at a distance bigger than the horizontal length of the shadow (e.g., about 10% bigger for ASTER or Landsat). In this work, we focus on recovering shadow-derived height differences from images acquired by scanning sensors, which is the more general case and can be easily expanded to non-scanning sensors.

The following sections describe the improved shadow-height methodology outlined above and present a robust validation in relevant areas, such as the mountains of the central Andes and Antarctica. We also explore other applications, such as ice shelf freeboard measurements and the study of volcanic plumes.

## 2. Materials and Methods

### 2.1. Used Imagery

To develop and validate the proposed shadow-height methodology, we used imagery acquired between 2000 and 2008 by the Advanced Spaceborne Thermal Emission and Reflection Radiometer (ASTER) onboard the Terra satellite [18]. The ASTER sensor provides along-track stereo pairs for each acquisition using nadir and backwards-looking cameras. This capability allowed us to perform preliminary validation of shadow-derived height differences using photogrammetric techniques on the stereo pairs. Additionally, ASTER imagery contain rich metadata, including a detailed record of the position of the satellite and the timing of the data acquisition. This information allowed us to focus on the novel elements of the methodology, leaving aside the complexities of orbital parameter determination and orbital solutions. We used ASTER L1A data products for the validation procedure, which we processed in Matlab using custom and inbuilt functions (see Supplementary Materials).

### 2.2. Problem Geometry

To avoid making assumptions about the geometry for the shadow-height problem and to present a methodology which is applicable to any platform, we need to consider the camera's movement during the image acquisition. We could neglect this additional complexity when the whole image is acquired simultaneously, such as in a standard digital or analogue camera with a mechanical shutter. However, this consideration is critical when we want to use scanning sensors. Whisk broom and push broom scanning sensors are the most common on Earth-observing imaging satellites. To tackle the most general case, we present a method, illustrated in Figure 3, consisting of the following steps.

1. Pin down the position of the projector **P** and the shadow **S** on the image (procedure detailed in Section 2.5).
2. Find the satellite position at projector observation time (**Sp**) and apparent projector position over the ellipsoid using internal satellite orientation parameters (**Pe**). Then, define the 3D line **Sp–Pe** connecting the satellite and the projector.
3. Find the satellite position at shadow observation time (**Ss**) and the apparent position of the shadow over the ellipsoid using internal satellite orientation parameters (**Se**). Then, define the 3D line **Ss–Se** connecting the satellite and the shadow.
4. Find the apparent position of the Sun, taking into account atmospheric refraction (⊙), at the time of observation.
5. Set the Projector position to use in the calculations (**P**).
6. Define the line ⊙–**P**, which is the line defined by the light ray coming from the center of the Sun that passes exactly by the tip of the projector toward the shadow.
7. Find the point of the line **Sp–Pe** closest to the line ⊙–**P** (**Pe′**) (in the ideal case, these lines intersect each other).
8. Find the point of **Ss–Se** closest to ⊙–**P** (**Se′**).
9. Find the distance **D** between **Pe′** and **Se′**.
10. Find the position of the shadow **S**, as the point on the line ⊙–**P** at a distance **D** from **P** in the direction opposite to the Sun.

We implemented astronomical and geometric functions to perform the above calculations in Matlab, which are included in the supplementary materials. The following sections describe the procedures we used to find the different positions mentioned in the previous ten steps.

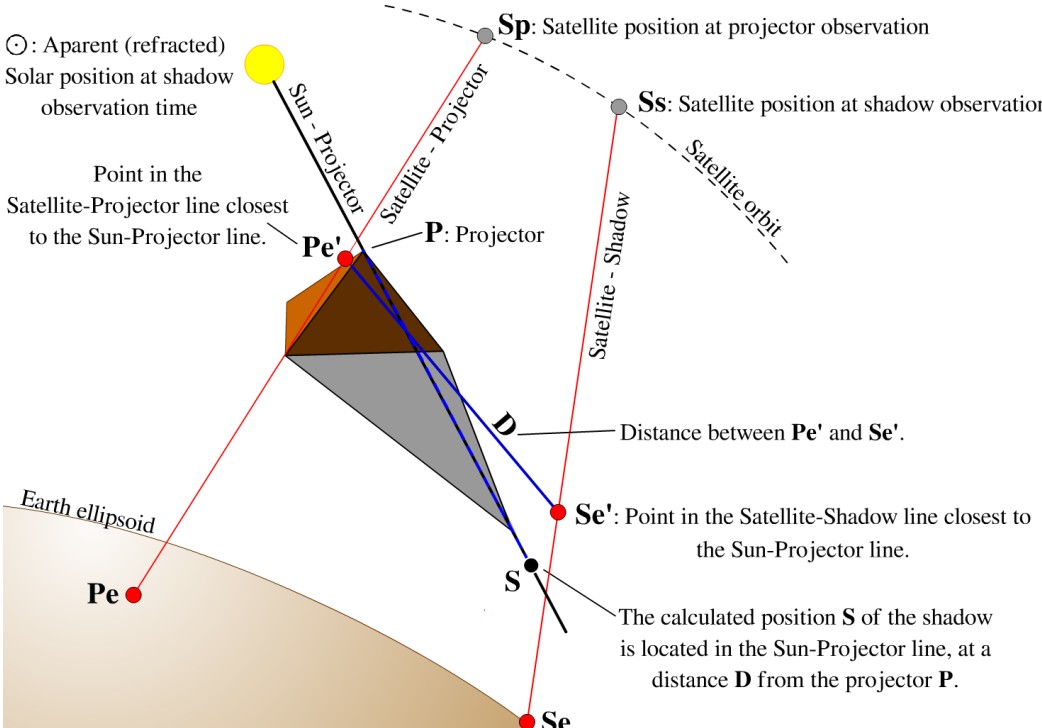

**Figure 3.** Schematics of the calculation process for the exact position of a shadow.

## 2.3. Satellite Position

Satellite positions could be calculated from orbital parameters at any given time. However, most Earth-observing satellites have orbital velocities of about 30,000 Km/h, thus requiring precision timing and updated ephemeris (i.e., orbital parameters) to achieve high positioning accuracy. Nonetheless, low-level satellite data products may include metadata with the exact acquisition time and satellite position for selected pixels. In the case of ASTER L1A data products, positions are included within the metadata at one meter resolution and times at one millisecond resolution for 132 lattice points evenly distributed across the image. This information allows for accurate satellite positioning at the acquisition time of each line of pixels within the image. We use this information and interpolation between lattice points to estimate the exact position of the satellite at the time of observation of the projector **Sp** and the shadow **Ss**.

## 2.4. Ellipsoidal Apparent Positions

The apparent ellipsoidal position of a point in the image is where the instantaneous visual from the satellite to the point intersects the ellipsoid. The displacement relative to the actual position of the point is proportional to its ellipsoidal height and the off-nadir observation angle. Errors in this position are mostly related to errors in the satellite orientation parameters. We calculate apparent ellipsoidal positions using the satellite positions **Sp** and **Ss** and the orientation parameters of the imaging platform. The ASTER L1A product includes apparent ellipsoidal positions at selected lattice points. We calculate the apparent ellipsoidal position of any point in the image by interpolation from those reference positions.

## 2.5. Shadow and Projector Picking

The precise determination of the position of the shadow and the projector on the image space are two crucial steps that are particularly prone to errors. We first perform preliminary selection of the image coordinates by manually clicking in the image at our best estimation of the positions of the shadow by visual inspection. Then, to achieve sub-pixel accuracy, we perform a fine adjustment by fitting a theoretical illumination profiles to the

image data. In some cases, the preliminary selection is straightforward; however, in others, it may be challenging to match a given projector with its corresponding shadow correctly. To help with this matching, after we select the first point, we plot theoretical shadow direction lines, either down from the projector or upwards from the shadow. Preliminary shadow positions will be used as well to evaluate the improvements achieved by the fine adjustment of the shadow positions.

The illumination profiles of shadows and projectors are different, and we describe each case independently in the following two sub-sections. For greater clarity, from now on, the term "elevation" will be exclusively used to refer to the solar angle above the horizon. Similarly, the term "height" will be exclusively used to refer to a vertical distance above the ellipsoid.

### 2.5.1. Shadow Calculation and Picking

The edges of shadows are diffuse; this is the result of light diffraction and the fact that the Sun is not a punctual light source. Instead, it has an angular diameter of about 0.5°. In the shadow-height method, the position of the Sun always refers to the position of its center. Therefore, when looking for the position of the shadow, we want the position where the shadow would be if all of the luminosity of the Sun were concentrated at its center. Thus, we call this point the "center" of the shadow. While the diffuse nature of shadow edges might seem an obstacle to accurately finding their center at first, it is actually an advantage that makes it possible to locate them with fine sub-pixel accuracy. The first step to exploit this advantage is to calculate the theoretical shadow illumination profile, which then needs to be fitted to the data. This theoretical illumination profile is affected by the luminosity distribution across the solar disk, the shape of the projector, and the geometry of the surface where the shadow is projected. Here, we describe how we estimated and combined these factors, starting with estimating the intensity distribution across the solar disk.

While the Sun might present sunspots and other luminosity irregularities, the most important effect to take into account is limb darkening. Limb darkening is produced by radiance variations in the solar atmosphere, resulting in the solar disk becoming darker towards the edges. Figure 4b shows a photographic image of the Sun, in which the limb darkening effect is visible.

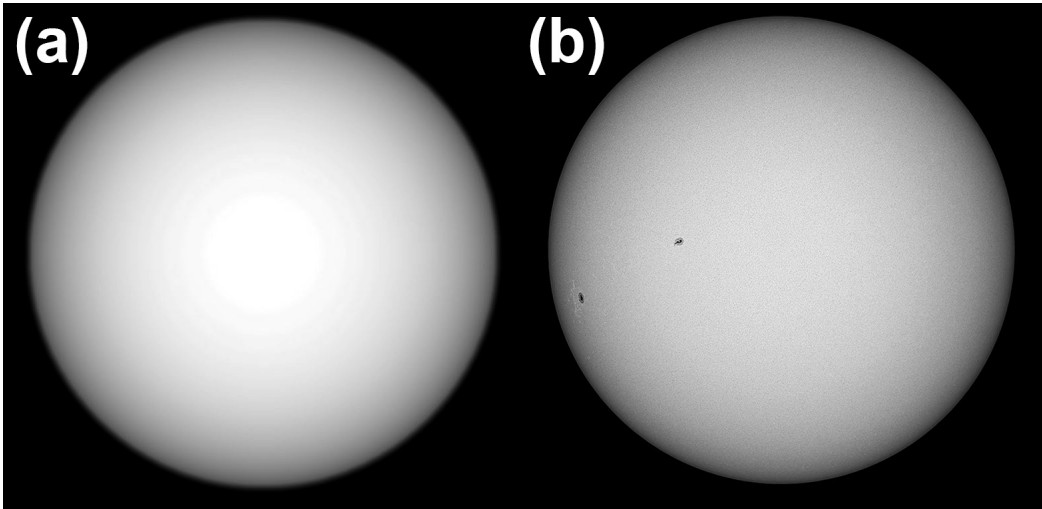

**Figure 4.** (**a**) Calculated solar disk intensity distribution. (**b**) Real image of the solar disk showing limb darkening (© Matúš Motlo).

We modeled limb darkening using the formulation presented by Cox [19], where the intensity of a point on the solar disk is given by

$$I = I_0 \sum_{k=0}^{n} a_k \left( \frac{\sqrt{\cos^2 \Theta - \cos^2 \Omega}}{\sin \Omega} \right)^k, \qquad (1)$$

where $\Omega$ is the solar angular radius (semidiameter), $\Theta$ is the angular distance from the center of the Sun to the point at which the intensity $I$ is calculated, $I_0$ is the central intensity, and $a_k$ are scalar coefficients. For the Sun at a wavelength of 550 nm, limb darkening is computed with $n = 2$ and

$$a_0 = 0.30, \qquad a_1 = 0.93, \qquad a_2 = -0.23 \qquad (2)$$

Another considered phenomenon affecting the apparent intensity distribution across the solar disk is the variations of the strength of light extinction through Earth's atmosphere. However, we neglected this effect, as it turned out to be negligible within the narrow angular span of the Sun ($\sim$0.5°).

Given that we require relative intensities only, we calculate a theoretical illumination distribution of the solar disk using Equation (1) with the coefficients in Equation (2), and an arbitrary value of $I_0 = 1$. Then, we normalize the illumination intensity values, such that the sum over the whole disk is equal to one. Figure 4a shows a calculated solar disk with a radius of 100 pixels.

The following step to calculate the theoretical illumination profile of a shadow involves estimating the shape of the projector. The relative intensity at each point along the illumination profile of a shadow is mainly a function of the solar elevation relative to the projector; however, the shape of the projector and the slope of the surface in the shadow direction can also significantly affect the illumination profile. The shape of the projector can be estimated by measuring the angle at the tip of the shadow (see Figure 5e). We use a simple approach by approximating the projector as a triangle and its shadow as an elongated triangle with the same base length. Then, the angle of the projector $\psi$ (see Figure 5f) is given by:

$$\psi = 2 \tan^{-1} \left( \frac{\tan(\psi'/2)}{\tan \theta} \right), \qquad (3)$$

where $\psi'$ is the apparent angle measured at the tip of the shadow (see Figure 5e) and $\theta$ is the solar elevation.

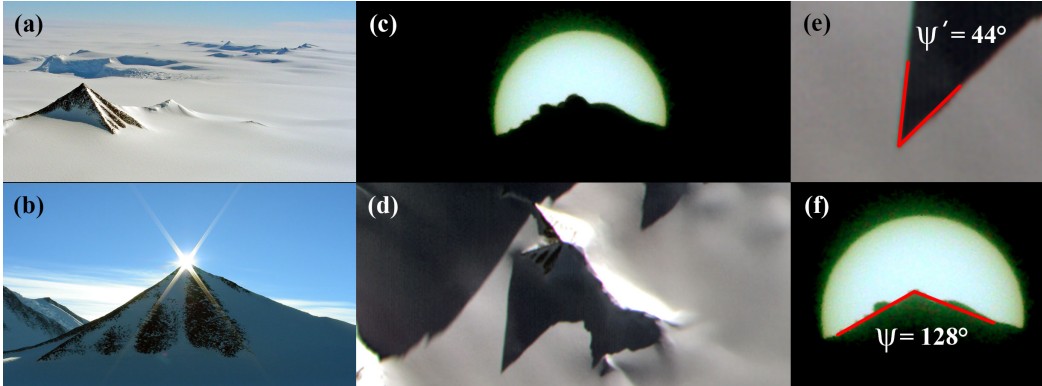

**Figure 5.** (**a**) Schatz Ridge Peak in the Sentinel Range, Antarctica (Latitude 78.5°S). This Peak was used as one of the validation points for this method. (**b**) Schatz Ridge Peak, as seen from the center of the shadow of its summit. (**c**) Close-up of the summit (projector) with a solar filter from the center of the projected shadow. Note that limb darkening is noticeable. (**d**) Schatz Ridge in a scene taken on 23 December 2004, with a Sun elevation of 12.4°. (**e**) Close-up of (**d**) with the value of the angle $\psi'$. (**f**) Close-up to (**c**) with the value for the angle $\psi$.

Figure 5 shows an example of this process performed over a pyramidal shaped peak in Antarctica. Using the values for $\psi'$ given in panel (e), the corresponding solar elevation of 12.4°, and Equation (3), the resulting value for the real angle $\psi$ is 123°, a reasonable approximation to the 128° measured in Figure 5f. The pictures in Figures 5b,c,f were taken on 5 December 2007, under similar solar elevation and azimuth as the ASTER scene in panels d and e, acquired on 23 December 2004.

To obtain the relative intensity received at any given point in the shadow illumination profile, we need to calculate how much of the solar disk is visible at that point. We do this by superimposing the simplified triangular projector over the calculated solar disk, based on the elevation of the solar disk over the projector, as seen from each point. Then, we estimate the intensity observed at the given point by integrating the pixel intensities of our modeled solar disk (such as in Figure 4a) that are not blocked by the projector. This process is illustrated in Figure 6.

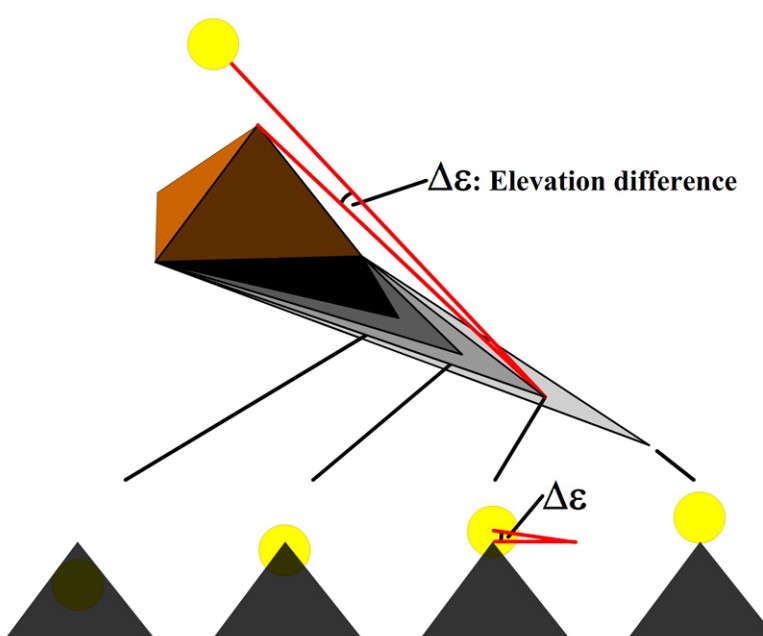

**Figure 6.** Schematic of the theoretical shadow profile calculation process. We compute the intensity at any given point using the solar elevation difference $\Delta\epsilon$ to superimpose the simplified triangular projector over the calculated solar disk. Then, we integrate the intensity within the fraction of the solar disk visible from any given point across the shadow edge.

The final step is to determine where in the image the theoretical illumination profile best fits the illumination gradient observed through the transition zone from the shade to the lit-up part of the image; however, the actual illumination profile is also affected by the slope and elevation of the surface where the shadow is projected, which is what we wish to calculate in the first place. To overcome this problem, we first compute the theoretical shadow illumination profile for a flat horizontal surface at an approximated altitude. This approximated altitude can be obtained using an approximated Digital Elevation Model (DEM) or from the shadow itself by iterating this method from an initial estimated altitude.

Figure 7 shows three theoretical shadow illumination profiles using different projector geometries, either considering or neglecting limb darkening. We can see how the profile shape varies significantly with the projector shape and is more sensitive to limb darkening for angular than flat projectors.

The fitting process can be done automatically; however, at the current level of development of the software tools used in this study, human supervision gives much better results. In the future, completely automated methods could be developed to compute multiple height differences along a shadow edge. The fitting process consists of superimposing

the intensity profile measured in the image with the theoretical one and moving the latter along the first to find the position that results in the best least-squares match. We also adjust three remaining unconstrained parameters during the fitting process to optimize the fit: Terrain inclination and minimum/maximum intensities. Once we complete the fitting process, we perform the fine adjustment of the preliminary shadow positions (identified by visual inspection) by setting the center of the best fit position of the theoretical profile as the center of the shadow. From this location in the shadow, the center of the Sun and the projector appear in the same direction ($\Delta\epsilon = 0$).

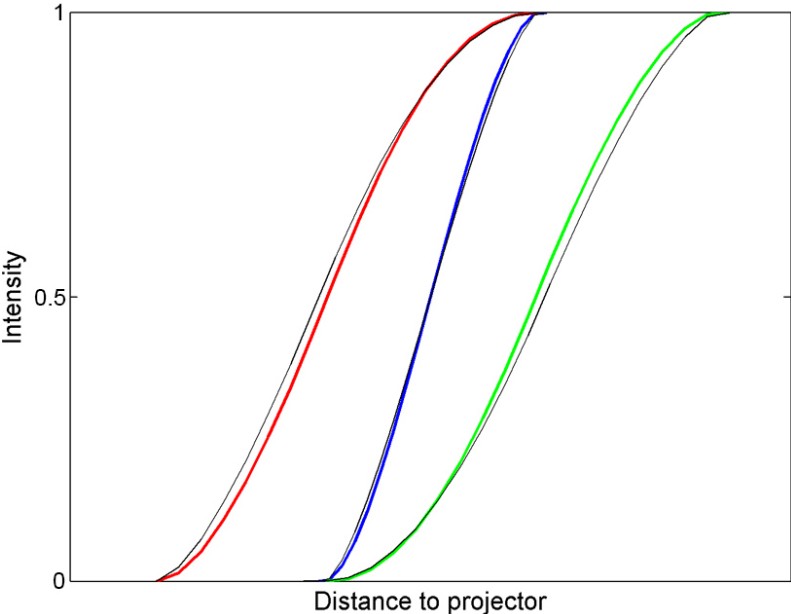

**Figure 7.** Theoretical shadow illumination profiles for different projector geometries: A 45° pyramidal Peak (red, $\psi = 45°$), a flat ridge (blue, $\psi = 0°$), and a 45° triangular notch (green, $\psi = 135°$). Black lines correspond to the same geometries, but neglecting limb darkening.

Shadow detection and fitting can be problematic for shadows projected onto surfaces with irregular or low reflectivity. The former is common in areas with discontinuous snow coverage. Surfaces with irregular reflectivity produce distortions in the intensity profiles, making the matching process more complex and less accurate. Water bodies are the most common low-reflectivity surface, which are often present around ice shelves and icebergs. The reflectivity of water is particularly low in the visual and infrared bands. Within this range, the maximum reflectivity occurs at the blue band. However, ASTER imagery does not have a blue band, making it virtually impossible to detect shadows over water.

### 2.5.2. Projector Selection

In this work, we focus on solid projectors with sharp edges, such as rocks or snow, as opposed to those with the soft edges, such as trees, vegetation, or clouds. The illumination profile on an image across a projector with sharp edges is simpler to model, and will often be composed of an illuminated pixel, a gray transition pixel, and a dark pixel. If there is no transition pixel, the projector position is well-known and can be located precisely between the illuminated and dark pixels. However, in most cases, there will be one or more gray pixels in between, and we use the intensities of these pixels relative to the bright and dark sides to constrain the sub-pixel position of the projector.

Figure 8 shows how we use the intensity of the gray transition pixel to pin down the sub-pixel position of the projector. In a real profile, the shadow direction is typically diagonal to the image's pixel grid; therefore, the intensity steps in the profile have different lengths, and there is more than one transition pixel. We opted for manual selection of the

projector positions by visual inspection of the profiles and the image; however, automation of this process would not be difficult to implement.

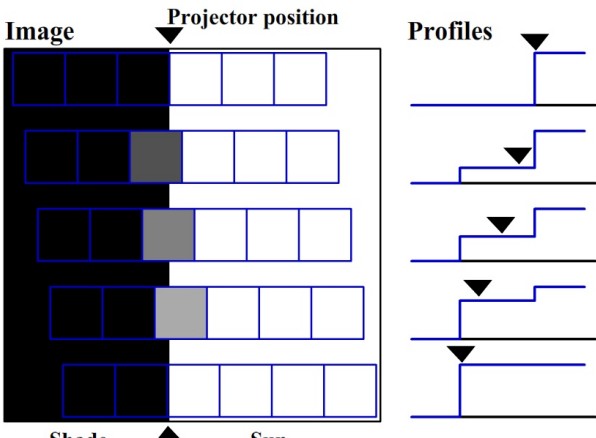

**Figure 8.** (**Left**) Simple image model with a projector along the center, where five different profiles are displayed. (**Right**) Intensity profiles for each case; we can see how the intensity of the transition pixel is related to the sub-pixel position of the projector (indicated by black triangles).

*2.6. Shadow Path Prediction*

In most cases, we are interested in neither prominent nor sharp projectors, making it confusing which shadow belongs to which projector. To overcome this ambiguity, we implemented an accurate shadow path prediction algorithm. A simple approach to this problem is to draw a line with the same azimuth as the Sun. However, such an approach can lead to errors up to hundreds of meters per kilometer of shadow length in off-nadir views. Our algorithm considers the problem geometry, satellite displacement during acquisition, and the curvature of light rays as they travel through the atmosphere, in order to avoid such errors.

*2.7. Apparent Position of the Sun*

The apparent position of the Sun corresponds to the position where the Sun appears to be to an observer on the Earth's surface. This position depends on the real position of the Sun and the total atmospheric refraction affecting the observer. Note that atmospheric refraction only affects the elevation of the Sun above the horizon, not its azimuth. The apparent solar position is, thus, computed as the point at the same distance and azimuth as the Sun, but with an elevation offset equal to the refraction angle at the position of the shadow. We separately describe the methodologies used to calculate the position of the Sun and the atmospheric refraction.

2.7.1. Real Position of the Sun

We calculate the position of the Sun using a simplified version of the VSOP87 theory described by Meeus [20], which is accurate up to one arcsecond (0.0003 degrees). This algorithm was adapted to provide positions in geocentric rectangular coordinates (in meters).

An input variable required in this calculation is the Terrestrial Time at the observation moment. The Terrestrial Time differs from the Universal Time (UT) by an empirically determined quantity, provided by the U.S. Naval Observatory at http://maia.usno.navy.mil (accessed on 15 October 2008).

2.7.2. Atmospheric Refraction

Atmospheric refraction can be calculated using many different approaches, some of which use standard atmospheric parameters, while others take into account pressure and temperature variations in a simplified single-layer atmosphere. Atmospheric refraction

algorithms are usually intended for astronomical applications, a field in which observing at low elevations is generally avoided due to atmospheric distortion and turbulence. Refraction at the horizon is generally modeled as a special case to determine the rise and set times of celestial objects. Therefore, the requirement of high-accuracy refraction estimations at elevations as low as 3–5° is uncommon. To increase the accuracy of this critical parameter, we chose a precise method that accounts for the atmospheric temperature, pressure, relative humidity, and vertical temperature lapse rate.

In order to avoid errors derived from the use of average values or rough estimations of meteorological variables, we used archived global atmospheric models at the National Oceanic and Atmospheric Administration (NOAA) to obtain the pressure, relative humidity, and vertical temperature profiles (lapse rate) for the location, date, and time of the image acquisition. Meteorological data are available from an increasing number of sources. We used the NOAA FLN archive [21] for dates before December 2004, and the Global Data Assimilation System one-degree (GDAS1) database [22] from that date onward. FLN data cover 1997 to 2006 at 6-h intervals, while the GDAS1 database covers from December 2004 to the present at 3-h intervals; both data sets have a spatial resolution of 1 degree.

The refraction calculation was carried out using the precise algorithm proposed by Kenneth [23], based on a physical model using quadrature. In this model, the atmosphere is assumed to be spherically symmetric, in hydrostatic equilibrium, and to obey the perfect gas law for a combined mixture of dry air and water vapor, as well as for the dry air and water vapor separately. It considers two atmospheric layers: The troposphere and the stratosphere. The troposphere extends upwards from the surface of the Earth to the tropopause, which corresponds to the boundary with the stratosphere and is assumed to be at a fixed altitude of 11 km. In the troposphere, the relative humidity is assumed to be constant and equal to its value at the position of the observer, and the temperature is assumed to decrease at a constant lapse rate. Kenneth [23] has used a fixed lapse rate value based on a standard atmosphere where the temperature drops at a rate of 0.0065 K/m. However, this value can vary widely in real conditions, especially in the Antarctic atmosphere, which deviates significantly from the "standard atmosphere". Therefore, for improved accuracy in non-standard atmospheric conditions, we discarded this assumption and considered the lapse rate as an input variable, which we obtained from the meteorological data described above.

Through the stratosphere, no pressure due to water vapor is considered, and the temperature is assumed to remain constant and equal to its value at the tropopause. Stratospheric refraction is neglected above an altitude of 80 km.

According to this model, the total bending of a ray is given by:

$$\epsilon = - \int_0^{z_0} \frac{r \frac{dn}{dr}}{n + r \frac{dn}{dr}},\tag{4}$$

where $z$ is the zenithal angle, $n$ is the refraction index, and $r$ is the distance to the center of the Earth. This integral is a transformation of the usual refraction integral, which makes it more suitable for numerical quadrature. The algorithm to solve this integral and to calculate $z$ and $n$ has been explained in detail by Kenneth [23].

### 2.8. Sensitivity to Errors in Input Variables

The accuracy of measured height differences depends on the errors of input variables. We review these error sources, in order to assess their magnitude under different scenarios and remote sensing platforms. The most relevant input variables and the factors affecting them are as follows:

- **Position of the projector**. The accuracy of the determination of the 3D position of the projector depends on the data source used. This source can be the image itself, a map, a reference image, or a field measurement.

- **Position of the satellite**. The accuracy, in this case, depends on how well the Ground Control Segment of the imaging platform tracks the spacecraft and the availability of such data in the ancillary information of the image.
- **Apparent solar position**. As discussed above, this value depends on the determination of the solar position and the atmospheric refraction which, in turn, depends on atmospheric parameters such as pressure, temperature, and relative humidity.
- **Shadow length measurement**. The accuracy, in this case, is affected by the internal deformation of the image and how well we can locate the projector and the shadow on the image.

Table 1 summarizes the a priori magnitude of the different error sources for the ASTER and a generic high-resolution spaceborne imaging platform. The estimated error ($\varepsilon$) has two components: An error bias and a relative error. The total error takes the form

$$\varepsilon = \beta \tan \theta + \alpha \Delta h, \tag{5}$$

where $\theta$ is the solar elevation angle, $\beta$ is the error bias proportionality constant, $\alpha$ is the slope of the relative error, and $\Delta h$ is the measured height difference. Considering height differences from 10 to 1000 m, the relative error component of Equation (5) ranges between 2 mm and 2 m using the estimate for $\alpha$ listed in Table 1 for the ASTER platform. Therefore, in most applications, this relative error will be much smaller than the error bias arising from the uncertainty in the size of the shadow. This bias is proportional to $\tan \theta$; thus, it will be large for high solar elevations and small for lower solar elevations. For example, using the estimated values for ASTER in Table 1, it would be around 14.25 m at a solar elevation of 45°, but only about 1.25 m at a solar elevation of 5°. Interestingly, for multiple measurements along the edge of a shadow (e.g., projected by a mountain ridge or the edge of an ice shelf), this error would arguably be mainly random. Therefore, averaging multiple measurements along a single shadow could considerably increase the accuracy of the height difference estimation. Another consideration regarding the total error estimated in Table 1 is that the worst geometry considered for some error sources would be favorable geometries for other sources. Therefore, the reported total error is likely to be overestimated. Through the rest of this section, we present more details on the characteristics of each error source and how we estimated their magnitudes.

**Table 1.** A priori error breakdown by source for two different platforms. Errors expressed as a percentage are relative to the absolute value of the measured height difference.

| Error Source | Upper Bound Magnitude for ASTER with No Field Measurements | Contribution to Calculated Height Difference Error | Upper Bound Magnitude for High-Resolution [1] Satellite with Field Measurements | Contribution to Calculated Height Difference Error |
|---|---|---|---|---|
| Projector 3D position | 100 m | 0.04% | 0.1 m [2] | 0.00% |
| Satellite positioning | 150 m | 0.02% | ~10 m | 0.004% |
| Size of the shadow | 0.95 pixels | $14.25 \tan \theta$ [m] [3] | 0.95 pixels | $0.95 \tan \theta$ [m] [3] |
| Atmospheric refraction | 0.7 arcminutes | 0.12% | 0.7 arcminutes | 0.12% |
| Total [4] | | $14.25 \tan \theta$ [m] + 0.18% | | $0.95 \tan \theta$ [m] + 0.16% |

[1] Assuming a ground resolution of one meter (pixel size). [2] Using dual-frequency differential GNSS. [3] Where $\theta$ is the apparent solar elevation. [4] We performed arithmetic instead of quadratic sum of errors, due to the relatively small number of error sources involved.

### 2.8.1. Sensitivity to Projector and Satellite Position Errors

To estimate the errors in projector and satellite positions, we conducted thousands of test calculations, introducing errors on input values over the whole range of possible geometric configurations. These errors turned out to be negligible or relatively small for spaceborne platforms. In the case of sun-synchronous satellites, such as Terra (the spacecraft carrying ASTER) or the Landsat series, the orbital altitude is above 600 km. As a

result, typical distances between the projector and the satellite are very large, compared with likely errors in the positioning of the projector.

In the case of the horizontal and vertical positioning of the projector, when retrieved from satellite imagery, DEMs, or maps, likely errors are typically in the 10–100 m range. For such errors, assuming the worst possible geometry (sun and satellite aligned at ~45° elevation), the relative error in the calculated elevation difference would be 0.004% to 0.04%, respectively. These errors are minor, compared with the remaining error sources. Therefore, precise projector positions are required only if the absolute position of the shadow is needed, as was the case for our validation process.

The accurate positioning of the satellite is more challenging than for the projector. Some image sources, such as ASTER, provide this information within the ancillary data. In the case of ASTER, the nominal spacecraft positioning accuracy ($3\sigma$) is 150 m [24]. In more recent platforms, the positioning accuracy can be down to 10 m or less; for example, for WorldView-1, it has been stated as being less than 7.6 m. For older space-born imagery, such as Corona, Hexagon, or ex-URSS declassified imagery, spacecraft location information might not exist or remain classified; however, this information can be recovered using ground control points, although such procedures could result in larger positioning errors. Nevertheless, even considering spacecraft positioning errors between 150 to 2000 m and the worst possible geometry, the resulting relative error on the calculated elevation difference would be between 0.02% and 0.3%, resulting in a relatively minor contribution to the overall error.

2.8.2. Sensitivity to Apparent Solar Position Errors

While modern celestial mechanics has made it possible to solve the position of celestial bodies with an accuracy of up to a few hundredths of an arcsecond [20], a much larger uncertainty comes from the determination of the atmospheric refraction. Atmospheric refraction consists of the bending of the solar light as it crosses the atmosphere. The result of this bending of the light rays is that the Sun appears to be at a higher elevation above the horizon than it is. Atmospheric refraction is more significant at lower solar elevations, such as those considered useful for the presented methodology, typically ranging from a few degrees to 45°.

Figure 9 shows the atmospheric refraction as a function of surface atmospheric pressure and temperature for different solar elevations. Each band represents the refraction values associated with temperatures ranging from −50 to +30 °C using standard relative humidity and temperature lapse rate values. We can see that, for the lower solar elevations used in shadow-height applications, the atmospheric refraction can reach values above 15 arcminutes. In addition to experiencing stronger atmospheric refraction, low solar elevations are also more sensitive to errors in its value. Figure 10 shows how, at low solar elevations, the error in shadow-derived height differences is more sensitive to errors in solar elevation. Neglecting the effect of atmospheric refraction in shadow-height applications would typically produce relative errors ranging from 1% to 8%.

When we consider atmospheric refraction, the error derived from uncertainties in apparent solar elevation is one order of magnitude smaller and primarily associated with the atmospheric refraction angle estimation. This remaining error will be, in turn, related to the uncertainties in pressure, humidity, temperature, and the refraction model itself. For example, with a solar elevation of 10° at an altitude of 2000 m, a major miscalculation of atmospheric variables (e.g., 50 mb in pressure and 20 °C in temperature) would lead to a 0.7 arcminute error in apparent solar elevation. Combined with an unfavorable geometry, this error would result in a relative height difference error of 0.12%.

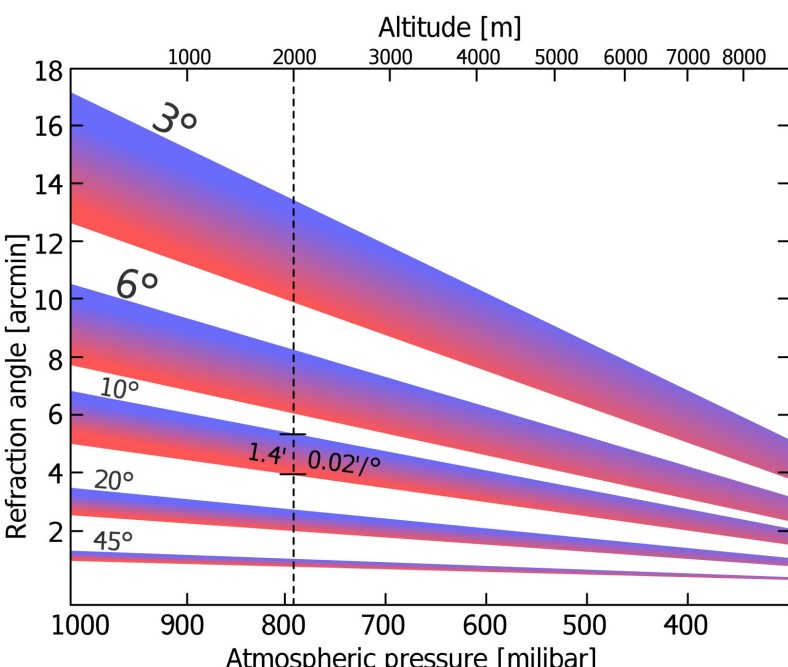

**Figure 9.** Atmospheric refraction as a function of surface atmospheric pressure for different solar elevations (3, 6, 10, 20, and 45 degrees). Each colored band represents a temperature range between −50 °C (upper blue border) and +30 °C (lower red border).

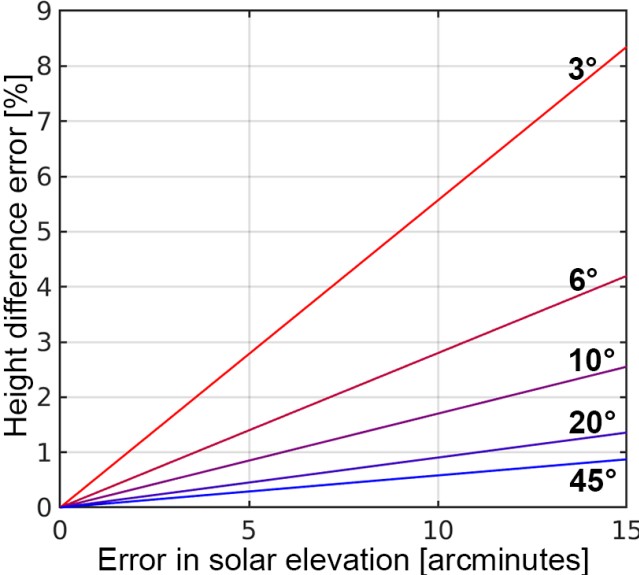

**Figure 10.** Relative error in shadow-derived height differences as a function of error in apparent solar elevation. Each line represents a different base value of solar elevation (3, 6, 10, 20, and 45 degrees).

### 2.8.3. Sensitivity to Shadow Length Errors

The methodology described above presents multiple techniques to improve the accuracy of identifying the correct start and endpoints of the shadow. From the methodology described in Section 2.5.2, we estimated our error in projector selection as equal to or smaller than 0.5 pixels. We can identify the position of the shadow with higher accuracy by taking advantage of the multi-pixel span of the shadow's edge and a modeled illumination profile, as described in Section 2.5.1. We estimated this error to be equal to or better than 0.25 pixels.

An additional source of error can arise from internal image consistency problems, such as jitter instability and telescope or spacecraft short-term orientation instability. These

problems result in pixels with varying angular sizes or pointing in inconsistent directions. The combined effect of these factors is carefully controlled, as they have a negative impact on the band-to-band registration process. This process is critical for a good matching between spectral bands. In the ASTER platform, this value is carefully controlled to ensure pixel-to-pixel matching better than ±0.2 pixels in any 8-min interval [24].

The total error considering uncertainties in the relative positioning of any point in the image and those associated with the selection of the beginning and end of the shadow was about ±0.95 pixels (0.5 + 0.25 + 0.2). We performed arithmetic, instead of quadratic, addition of errors, due to the small number of components involved. Therefore, for a pixel size of 15 m (as in ASTER images), the error bias constant will be given by $\beta = 0.95 \times 15 = 14.25$ m.

The influence of this shadow length error over the measured height difference depends on the problem geometry; in particular, the solar elevation $\theta$, the viewing angle of the satellite, and the slope of the terrain where the shadow is projected. However, for a first-order approach of the magnitude of this error, we can consider the basic geometry shown in Figure 1, assuming a nadir-looking image. In this case, for ASTER images, the error will be of 14.25 m for $\theta = 45^o$, 8.23 m for $\theta = 30^o$, 3.82 m for $\theta = 15^o$, and 1.25 m for $\theta = 5^o$.

*2.9. Validation Procedure*

To validate the proposed shadow-height methodology, we compared our shadow-derived height differences against independent measurements. While shadow-derived results could be verified using the stereo capabilities of ASTER imagery, we decided to instead perform validation against a higher-accuracy method, such as differential GNSS or laser altimetry by the Ice, Cloud, and land Elevation Satellite (ICESat). Nevertheless, we did use low-accuracy stereographic measurements during the development of the described methodology, which took place before the field measurements.

We performed validation measurements in three different areas, all located in the southern hemisphere and covering most of the latitudinal range where this methodology is applicable. The first two sites are mountain ranges, where the validation was made against on-site GNSS measurements. These sites are:

- **Sentinel Range, Antarctica**. At a latitude of 75°30′ South and a base altitude of ∼2100 m. At this location, we measured nine shadows belonging to five mountain peaks.
- **Andes Cordillera, Argentina**: At a latitude of 32°50′ South and a base elevation of ∼2500 m. At this location, we measured nine shadows belonging to three mountain peaks.

For these sites, the validation procedure consisted of the following six steps:

1. Identifying an available ASTER L1A scene with date as close as possible to the planned fieldwork and with shadows suitable for measurement and validation. These are shadows cast over a reasonably flat area with homogeneous reflectivity, such as valley bottoms covered by glaciers or snow. Note that ASTER imagery became available to users at no cost in 2016. Therefore, we did not have access to the full ASTER archive when we performed the field measurements.
2. Measuring the positions of the peaks selected as projectors using a high-precision GNSS receiver placed at the summit.
3. Calculating the absolute positions (altitudes and horizontal coordinates) of the shadows observed in each selected ASTER image using our shadow-height methodology.
4. Reaching the calculated shadow positions on foot using a handled GNSS navigator. At these approximated locations, we performed high-accuracy dual-frequency GNSS measurements.
5. After post-processing the GNSS data, we calculated the horizontal differences between the estimated shadow position and the measured point. These differences were typically in the range of 1–4 m, due to errors in the handheld GPS positioning. Then, we used these differences (distance and azimuth) in the field to estimate the altitude differences between the estimated positions of the shadows and the measured points.

6. Finally, we assume the real shadow-projector height difference to be equal to the difference between the GNSS solutions of the projector and the shadow, and any discrepancy with the shadow-derived value was assumed to correspond to the error of the shadow-height method.

We measured GNSS positions using a Trimble 5700 dual-frequency receiver and post-processed them with two or more long-baseline services, such as AUSPOS [25], SCOUT [26], Auto GIPSY [27], and/or OPUS [28]. We compared output positions from these services to check solution stability. The final positions had estimated errors below 0.40 m.

The third validation area was the Abbot ice shelf in Antarctica, at a latitude of 72°12′ South and a base height of 0 m. We were not able to perform field measurements in this area. Instead, we compared shadow-derived results with ICESat laser altimetry data, which have been successful applied to the measurement of freeboard heights, e.g., [29–32]. In this area, the validation procedure consisted of the following four steps:

1. Identifying an ASTER L1A scene with shadows suitable for measurement and ICESat tracks with dates as close as possible to the image. Shadows are suitable for measurement when cast over high-reflectivity surfaces, such as sea ice or an ice melange.
2. Measure the ice shelf freeboard height along the ICESat track. This height corresponds to the difference between the average elevations of the last three points on the ice shelf and as many points as possible on the sea ice or ice melange.
3. Calculating the shadow-derived freeboard height along a short section of the edge of the ice shelf, perpendicular and intersecting the ground track of ICESat.
4. We assume the error in the methodology to be equal or less than the difference between the shadow and ICESat-derived measurements of freeboard height.

In addition to these validation sites, we present an applied example on a volcanic plume from Shiveluch Volcano in Kamchatka, Russia (56°39′12′′N, 161°21′42′′E). Figure 11 shows the used image (see Table 2 for scene details). The base height of the volcano is about 300 m. We computed shadow-derived elevation differences on five shadows cast by its plume, both in the nadir and backward-looking bands.

**Table 2.** ASTER L1A v.3 scenes used for validation.

| Scene ID | Date | Solar Elevation | Area |
|---|---|---|---|
| SC:AST_L1A.003:2006547875 | 18 April 2000 | 33.3° | Andes |
| SC:AST_L1A.003:2012745748 | 9 April 2003 | 42.2° | Andes |
| SC:AST_L1A.003:2016330216 | 15 August 2003 | 36.1° | Andes |
| SC:AST_L1A.003:2022604196 | 11 April 2004 | 41.3° | Andes |
| SC:AST_L1A.003:2010581107 | 9 January 2003 | 24.5° | Sentinel Range |
| SC:AST_L1A.003:2027142602 | 23 December 2004 | 12.4° | Sentinel Range |
| SC:AST_L1A.003:2027171920 | 25 December 2004 | 27.4° | Sentinel Range |
| SC:AST_L1A.003:2032500227 | 4 January 2006 | 26.5° | Sentinel Range |
| SC:AST_L1A.003:2005797088 | 14 January 2002 | 3.7° | Abbot Ice shelf |
| SC:AST_L1A.003:2027212156 | 4 January 2005 | 5.1° | Abbot Ice shelf |
| SC:AST_L1A.003:2071551585 | 17 February 2008 | 19.8° | Shiveluch Volcano |

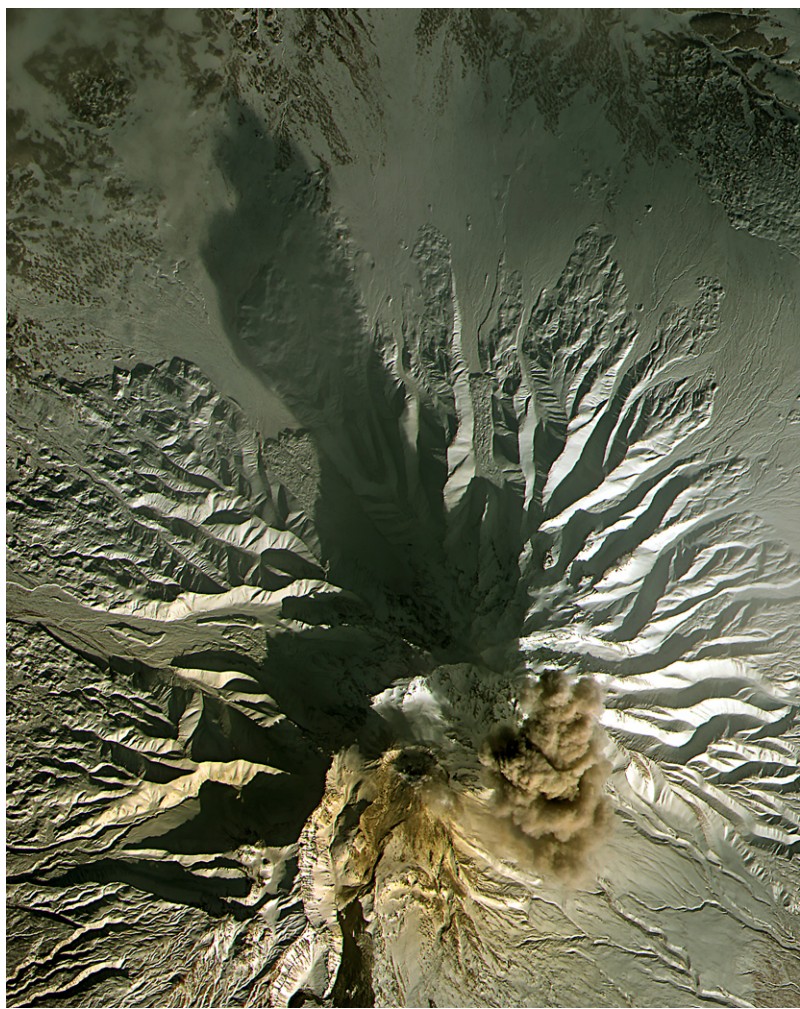

**Figure 11.** Detail of an scene by the Advanced Spaceborne Thermal Emission and Reflection Radiometer (ASTER) acquired on 17 February 2008. We applied the the shadow-height methodology to calculate the height and vertical velocity of an ash plume at Shiveluch Volcano in Kamchatka, Russia.

*2.10. Andes Validation Data Set*

The Andes validation site is located along the international highway connecting the city of Los Andes in Chile with Mendoza in Argentina. We measured validation points during September and October 2008. GNSS data acquisitions at each point lasted at least two hours.

The four ASTER L1A (v.3) scenes used (see Table 2) were acquired between 2000 and 2004, with solar elevations between 33.3° and 42.2°. We selected images both with and without snow cover. Snow depth at image acquisition time was estimated based on the extent of the snow coverage.

We measured nine shadows belonging to three peaks. The peaks selected as projectors were Mario Ardito, a sub-peak of Cerro Tolosa, and a sub-peak of Cerro Agua Salada, with elevations between 3199 m and 3863 m. We measured height differences ranging from 308 to 656 m. We summarize the measured coordinates and elevations of these peaks in Table 3. Figure 12 shows one of the scenes used for validation (11 April 2004), including the positions of the three peaks used as projectors and the two shadows measured in the scene.

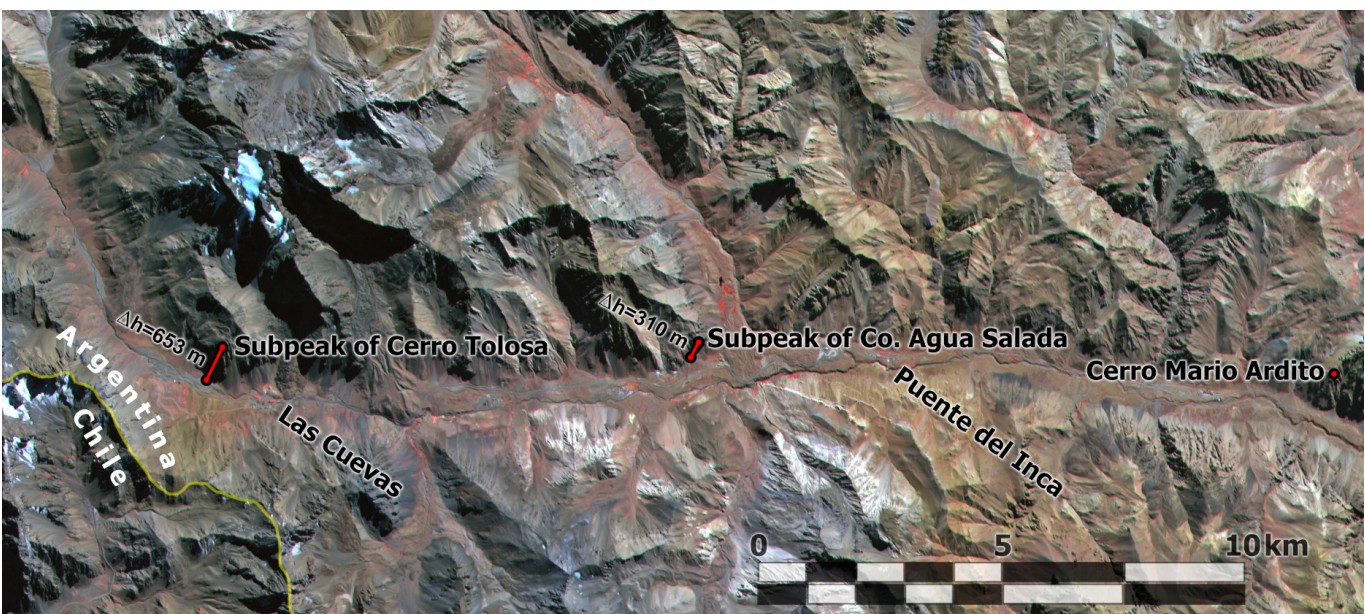

**Figure 12.** ASTER scene from 11 April 2004, used at the Andes validation site. Solar elevation for this scene was 41.3°, and the calculated refraction was 0.8 arcminutes. Red dots show all the features used as projectors in all four scenes at this validation site, and red lines show shadows used for validation in this scene. Next to each red line, the height differences between the shadow and projector are shown. For reference, we also show the position of the towns of Las Cuevas, Puente del Inca, and the Chile–Argentina border.

**Table 3.** Location of peaks selected as projectors for the Andes and Sentinel Range validation sites. Values derived from field GNSS measurements.

| Name | Latitude [°] | Longitude [°] | Elevation [m] | Area |
|---|---|---|---|---|
| Cerro Mario Ardito | −32.8416325 | −69.8130563 | 3199.43 | Andes |
| Sub-peak of Cerro Tolosa | −32.8019409 | −70.0564407 | 3863.28 | Andes |
| Sub-peak of Co. Agua Salada | −32.8157170 | −69.9513586 | 3296.58 | Andes |
| Mount Ryan | −78.369987 | −86.024942 | 3808.06 | Sentinel R. |
| Mount Ryan Southwest | −78.374103 | −86.032424 | 3769.77 | Sentinel R. |
| Schatz Ridge Peak | −78.481736 | −86.031342 | 2633.93 | Sentinel R. |
| Knutsen West ridge peak | −78.499830 | −86.044716 | 2488.70 | Sentinel R. |
| Gardner West ridge peak | −78.402533 | −86.190158 | 2540.63 | Sentinel R. |

## 2.11. Sentinel Range Validation Data Set

The Sentinel Range is located in the Antarctic interior and hosts the highest Antarctic mountains. The validation points in this area were measured as part of the sixth mapping expedition of The Omega Foundation to the Sentinel Range from November 2007 to January 2008.

The five peaks we used for validation are listed in Table 3. The elevations of their summits ranged from 2488 m to 3808 m, and the absolute height differences of the shadows were between 292 and 1495 m. Figure 13 shows the area and the locations of these five peaks, as well as two of the measured shadows. GNSS data acquisition at each point (peak or shadow) lasted for at least one hour, although we measured most of the points for more than 10 h. The only exception was Mount Ryan SW, which we measured for 35 min.

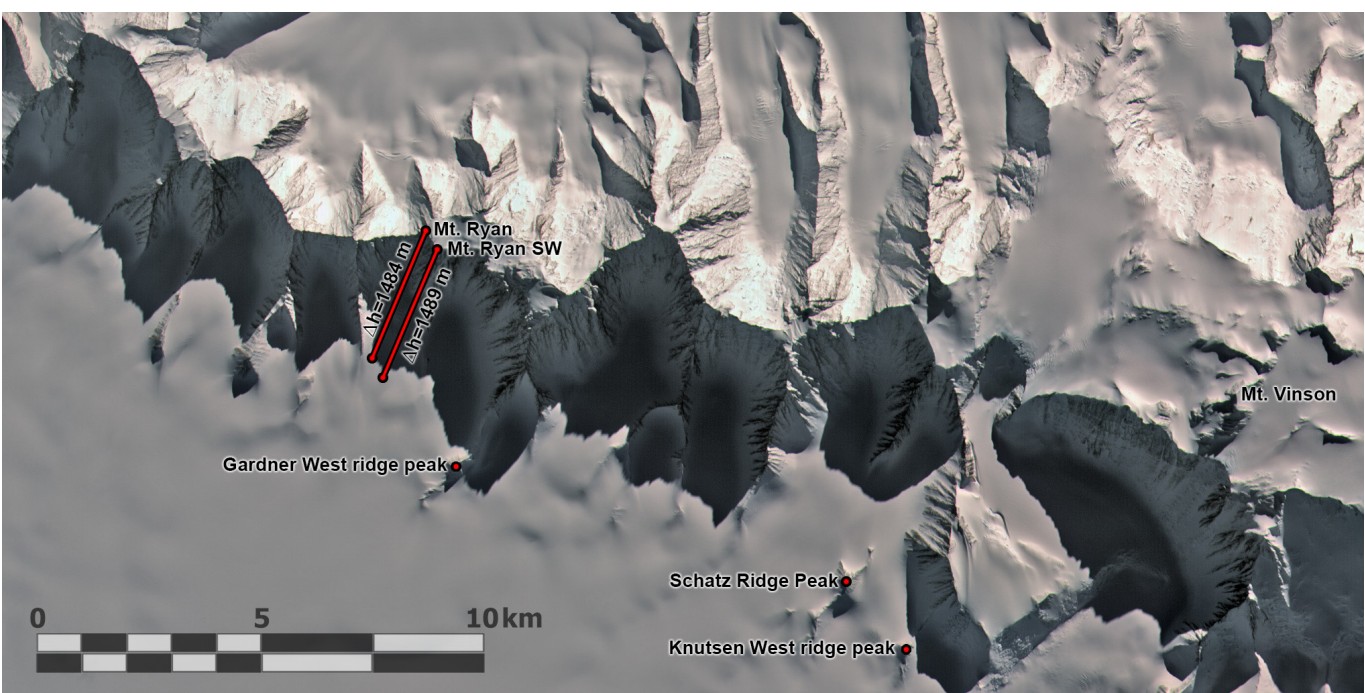

**Figure 13.** ASTER scene from 9 January 2003 used for the Sentinel Range validation site. Solar elevation for this scene was 24.5°, and the calculated refraction was 1.7 arcminutes. Red dots show all the features used as projectors in all four scenes used at this site. Red lines show shadows used for validation in this scene. Next to the red lines, the elevation differences between shadow and projector are shown. For reference, we also show the position of Mount Vinson, the highest mountain in Antarctica.

We selected four ASTER L1A (v.3) scenes (see Table 2), acquired between January 2003 and January 2006, with solar elevations between 12.4° and 27.4°. Figure 13 shows the scene from January 2003.

### 2.12. Abbot Ice Shelf Validation Dataset

We validated the presented methodology for measuring ice shelf freeboard heights on the Abbot ice shelf in Antarctica. Due to the low reflectivity of seawater on the bands available in ASTER data (which lacks a blue band), we selected two scenes displaying shadows projected over sea ice (see Table 2). The acquisition dates were January 2002 and January 2005, with solar elevations of 3.7° and 5.1°, respectively.

As reference elevation data, we used ICESat lidar profiles—in particular, the GLA12 v.28 product [33], for which seven continuous profiles over two different ground tracks were available. Figure 14 shows the Abbot ice shelf, ICESat ground tracks, and the two validation zones used.

We calculated reference freeboard heights (thickness of emerged ice) from ICESat height profiles by averaging several values over the sea to set base height, and the first three values over the ice sheet to set ice shelf surface height. Then, the freeboard height corresponds to the difference between base and surface heights. Note that freeboard heights are not sensitive to tidal variations, as the ice shelf is afloat.

Figure 15 shows the ICESat profiles over Zone 2, where variations in base and surface heights are most likely due to snow accumulation, ablation, and tides. Points with more significant deviations from the mean base height could correspond to small icebergs embedded in the sea ice or ice melange. Note that below-zero elevations are due to the difference between geoidal and ellipsoidal heights in the area.

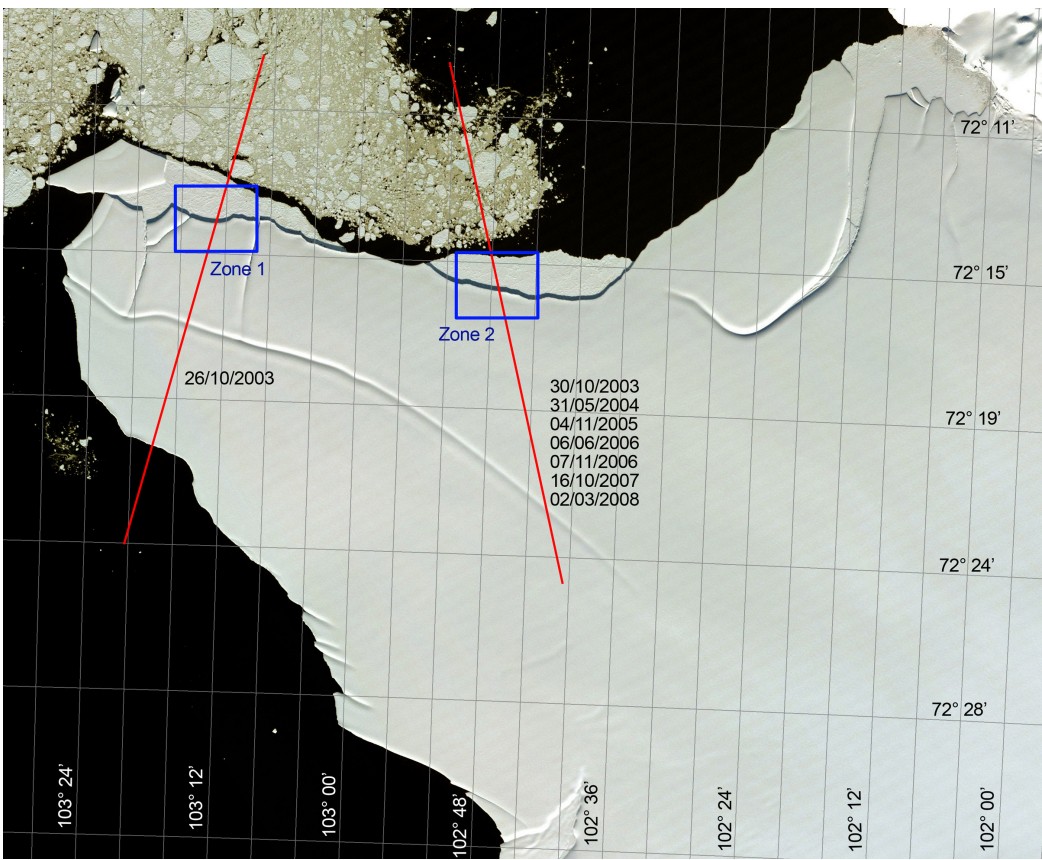

**Figure 14.** ASTER L1A scene from 4 January 2005, used for the Abbot ice shelf validation site. ICESat ground tracks are shown in red, together with the dates of the available profiles on each track. Validation measurements were carried out on each ground track using shadows projected over the sea ice, in the areas designated as Zone 1 and Zone 2 (blue boxes).

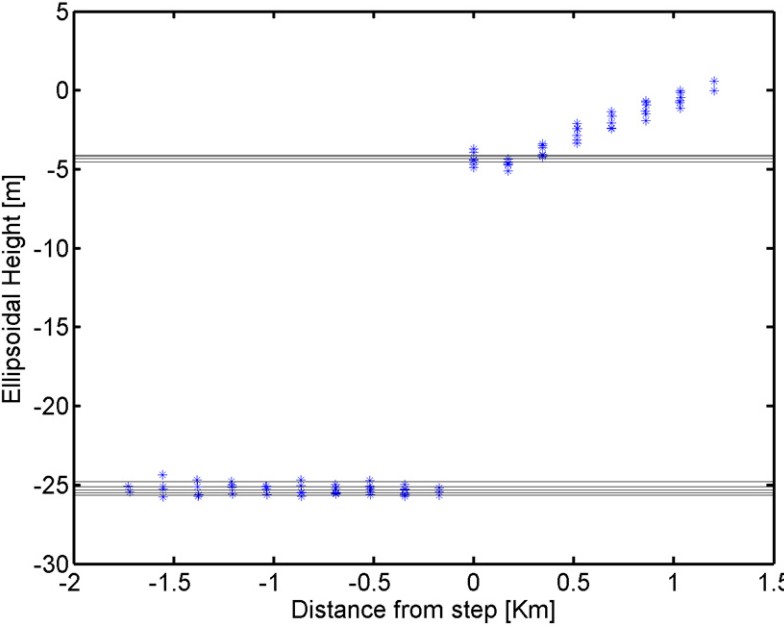

**Figure 15.** Available ICESat elevation profiles over Zone 2 (seven profiles). Horizontal distance refers to the first point over the ice shelf on each profile. Calculated surface and base heights for each profile are shown as black horizontal lines.

## 3. Results

### 3.1. Elevation Differences on Mountain Ranges

We computed shadow-derived height differences for the nine shadows selected for each mountain validation site. The calculation was carried out both in the nadir and backward-looking bands of the ASTER sensor, resulting in 36 independent measurements. Then, we compared each resulting height difference with high-precision GNSS solutions. Figure 16 shows the discrepancy between height differences measured from shadows and GNSS data. We can see that measurement errors were always below 9 m and the standard deviation of all measurement errors was 3.2 m for the Andes site and 3.9 m at the Sentinel Range site. In relative terms, all errors were below 2%. For comparison, thin markers on Figure 16 show the errors associated to height differences computed neglecting atmospheric refraction (black markers) or the fine adjustment of shadow positions (red markers).

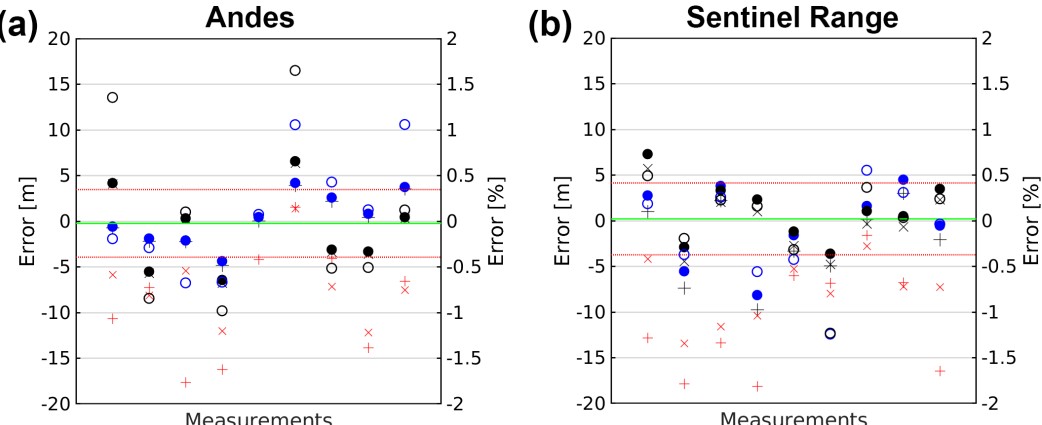

**Figure 16.** Errors in shadow-derived heights measurements at the Andes (**a**) and Sentinel Range (**b**) validation sites. Errors correspond to the difference between shadow-derived and GNSS measurements. Circles represent measurements done with the presented methodology, solid circles corresponds to the error in meters (left axis) and empty circles to the relative error (right axis). Blue and black circles represent nadir- and backward-looking bands respectively. The solid green line shows the mean value of errors, and the red dotted line shows the standard deviation of errors around the mean. Thin markers ("+" and "×" symbols) represent solutions neglecting atmospheric refraction (black) or fine shadow adjustment (red), in this case nadir- and backward-looking bands are represented by "+" and "×" symbols respectively.

These errors correspond to upper limits of the actual errors of our methodology, as they include all other errors within the validation process not attributable to the presented methodology. One of such error sources is terrain changes between field measurements and image acquisition. The snowfields and glaciers over which most of the shadows were projected suffer from variations in surface elevation. Fortunately, snow accumulation and ablation rates are minimal in the Sentinel Range area. In the case of the Andes site, all shadows were projected on bare ground or seasonal snow which, in the area, reach maximum snow depths of only a few meters.

We estimated the surface elevation rate of change at the Sentinel Range by repeating a measurement on the glacier below Mount Ryan taken two years earlier. Within this period, the elevation change was 0.63 m upwards. At the Andes site, the approximate snow depth at image acquisition time was estimated based on snow coverage for images with snow cover. We believe the error in those estimates to be less than one meter, based on knowledge of the area and typical snow depth values. Therefore, the errors shown in Figure 16 are likely to be slightly overestimated.

### 3.2. Ice Shelf Freeboard Measurements

We calculated Abbot ice shelf freeboard heights at multiple points along the ice shelf margin close to the ICESat tracks. Figure 17 shows details of Zone 2 on the two ASTER scenes used for validation at this site. Measured shadows are shown as black dots and the corresponding projectors as brown triangles. Shadow-derived measurements were calculated using imagery from 2002 and 2005, and reference ICESat data were acquired between 2003 and 2008.

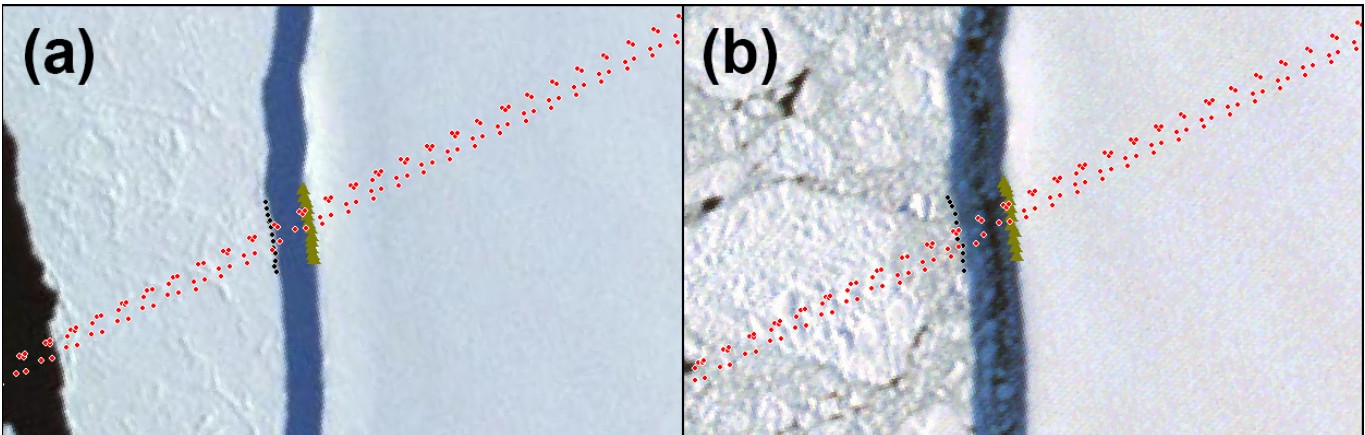

**Figure 17.** Detail of Abbot ice shelf Zone 2. Red dots show the center points of ICESat laser altimetry footprints used as reference freeboard height measurement. The points on the ice shelf margin chosen as shadow projectors are shown as brown triangles, and their respective shadows are shown as black dots: (**a**) ASTER scene from January 2005 (solar elevation of 5.1° and calculated refraction of 9.6 arcminutes); and (**b**) ASTER scene from January 2002 (solar elevation of 3.7° and calculated refraction of 12.5 arcminutes).

For Zone 1, only one ICESat elevation profile was available. It was acquired in October 2003 (see Figure 14), and its analysis resulted in a freeboard height of 21.69 m. For Zone 2, seven profiles were available between 2003 and 2008. Calculated freeboard heights range from 20.46 m to 21.33 m (0.87 m maximum difference), with a standard deviation of 0.30 m and a mean value of 21.01 m. We found no temporal or spatial trends; therefore, we assumed a constant freeboard height of 21 m for the two ASTER scenes used. This assumption probably led to a minor overestimation of errors (i.e., the method accuracy would be slightly better than reported).

Figure 18 shows the differences between shadow-derived freeboard heights and the reference heights calculated from ICESat data. Zone 1 shadows were measured only for the January 2005 image, as there was no continuous sea ice present in the January 2002 image for that area. Note that ASTER has no blue band and that seawater reflectivity is too low in the available bands to distinguish shadow edges adequately. In the January 2005 image, we measured ten shadows in Zone 1 over a 370 m section of the ice shelf edge around the ICESat ground track. The mean value of the difference was −1.46 m, with a standard deviation of 0.26 m, considering measurements in both the nadir and backward-looking bands (20 independent measurements in total). The difference between the maximum and minimum values was 1.2 m.

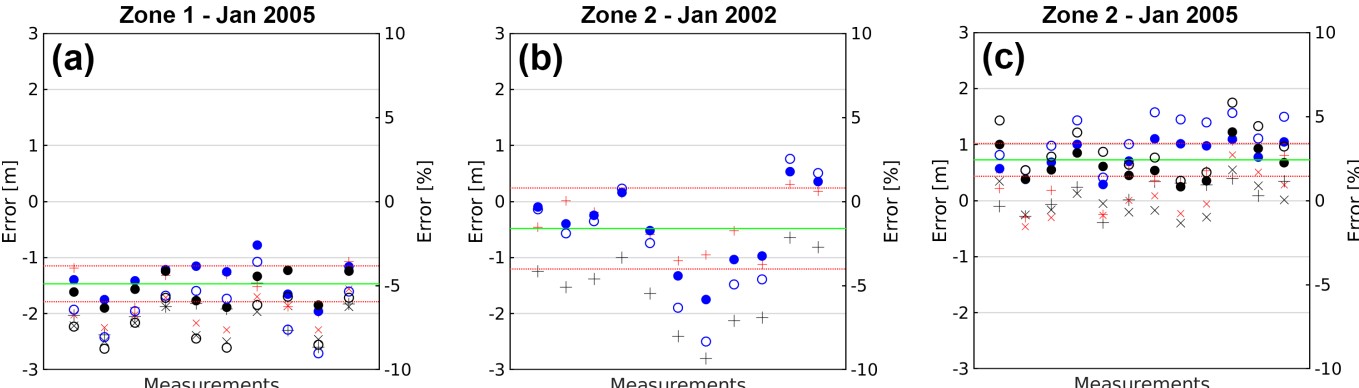

**Figure 18.** Errors of the shadow-derived freeboard heights measured at the Abbot ice shelf. Errors were estimated as the difference between shadow-derived measurements and reference values obtained from ICESat laser altimetry. Circles represent measurements done with the presented methodology, solid circles corresponds to the error in meters (left axis) and empty circles to the relative error (right axis). Blue and black circles represent nadir- and backward-looking bands respectively. The solid green line shows the mean value of errors, and the red dotted line shows the standard deviation of errors around the mean. Thin markers ("+" and "×" symbols) represent solutions neglecting atmospheric refraction (black) or fine shadow adjustment (red), in this case nadir- and backward-looking bands are represented by "+" and "×" symbols respectively. (**a**) Errors for 20 measurements carried out in Zone 1 on the ASTER image from 4 January 2005; (**b**) Errors for 11 measurements carried out in Zone 2 on the ASTER image from 14 January 2002; and (**c**) Errors for 24 measurements carried out in Zone 2 on the ASTER image from 4 January 2005. Zone locations are shown in Figure 14.

For Zone 2, seven ICESat profiles were available, which spanned a band 140 m wide, in which we estimated an average freeboard height of 21.01 m. Figure 18b shows the errors over Zone 2 for the January 2005 ASTER scene. The mean error was 0.73 m, with a standard deviation of 0.22 m. The difference between the maximum and minimum measured values was 0.98 m. For the January 2002 scene, Zone 2 was visible only in the nadir-looking bands. The average error was −0.48 m, with a standard deviation of 0.72 m. The difference between the maximum and minimum measured values was 2.3 m.

For comparison, Figure 18 shows also the errors associated to height differences computed neglecting atmospheric refraction (black markers) or the fine adjustment of shadow positions (red markers).

### 3.3. Shiveluch Volcano Plume Measurements

As an applied example beyond the cryospheric sciences, we carried out shadow-height measurements of a volcanic ash plume. Figure 11 shows details of the selected ASTER scene, portraying the plume resulting from the 17 February 2008 eruption of Shiveluch volcano. We calculated height differences between five plume features and their shadows from this image and, to establish the base level of the plume, we used a DEM by the Shuttle Radar Topography Mission (SRTM) [34]. Note that a known base level is required to compute the absolute elevation of the measured features.

The validation of these measurements was impossible, as we did not find any independent measurements. Additionally, the use of the stereo capability of ASTER was hindered by the significant displacement (∼400 m) of plume features during the 55.4 s elapsed between nadir- and backward-looking acquisitions. Nonetheless, assuming that errors would be similar to those observed in our validation areas, the shadow-derived elevation of volcanic ash plumes could produce plume heights with accuracy suitable for this particular application.

Interestingly, shadow-derived measurements on along-track stereo pairs can be used to estimate vertical plume velocities if a specific feature is identifiable in both images.

For ASTER images, the typical time delay between normal and backwards acquisition is slightly under a minute. As an example of how such measurements could work, Figure 19 shows close-up views of the Shiveluch plume and its corresponding shadow. We identified five features in both the nadir- and backward-looking bands by visual inspection. The availability of identifiable plume features depends on the characteristics of the plume; for example, very diffuse plumes are not suitable for shadow-derived height estimation.

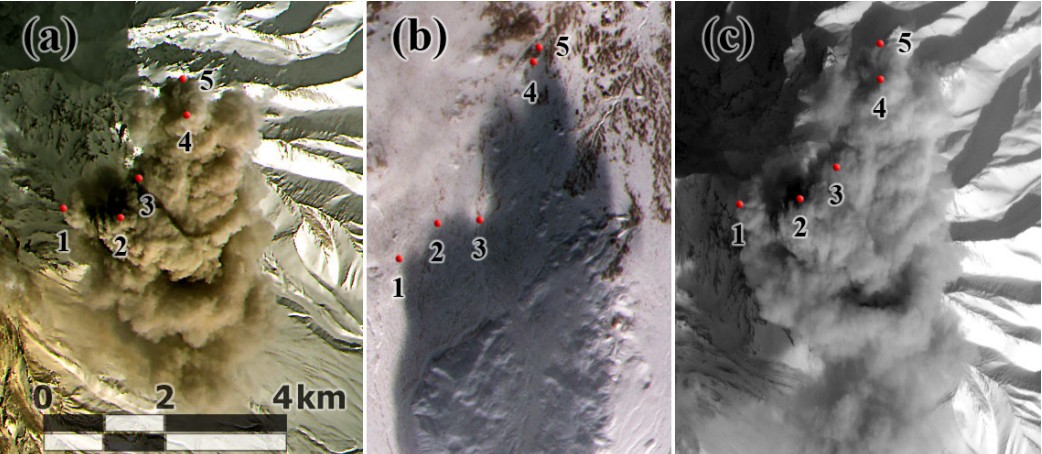

**Figure 19.** Shiveluch volcano ash plume features identified and measured in the nadir- and backward-looking bands. The time elapsed between nadir and backwards acquisitions was 55.4 s: (**a**) Features on the nadir-looking bands; (**b**) Corresponding shadows on the nadir-looking bands; and (**c**) Features on the backward-looking band.

We computed shadow-derived altitudes and positions for nadir- and backward-looking bands for the five features shown in Figure 19. We performed projector and shadow selection, with no significant problems, using the presented methodology. However, some difficulties arose due to reflectance irregularities in the shadow projection areas. Then, we computed horizontal and vertical velocities of the identified plume features using the shadow-derived positions. Table 4 shows the heights of the identified plume features and their vertical velocities during the time elapsed between nadir- and backward-looking acquisitions.

**Table 4.** Heights and vertical velocities of the features identified in the ash plume of Shiveluch volcano. The time elapsed between nadir- and backward-looking acquisitions was 55.4 s.

| Feature Number | Elevation Nadir-Looking | Elevation Backward-Looking | Vertical Velocity |
| :---: | :---: | :---: | :---: |
| 1 | 5.822 m | 5.740 m | −1.48 m/s |
| 2 | 6.214 m | 6.162 m | −0.96 m/s |
| 3 | 6.046 m | 6.161 m | +2.07 m/s |
| 4 | 6.690 m | 6.613 m | −1.39 m/s |
| 5 | 6.584 m | 6.465 m | −2.14 m/s |

## 4. Discussion

Several error sources have an impact on the presented shadow-height methodology. From Table 1, we can see that the two most significant error sources are the measured length of the shadow and atmospheric refraction.

Some previous studies using the shadow-height method have neglected the fact that the Sun is an extended light source, by assuming that the shadow edges are sharp, (see, e.g., [4,6,7,10,35]). While this is a reasonable assumption for small objects under high solar elevations, it can introduce significant errors in other cases. In reality, the angular size of the Sun is typically around 0.5°; which, by itself, results in an illumination gradient

at the shadow edge that can extend over several pixels on the image. The exact extent depends on the geometry but, assuming the simple geometry depicted in Figure 1, it would depend only on the solar elevation ($\theta$) and the measured height difference ($\Delta h$). For typical conditions at our validation sites, the illumination gradient would spread over the following approximate distances: 8 m for the Andes site ($\Delta h = 400$ m, $\theta = 40°$), 75 m for the Sentinel Range site ($\Delta h = 1000$ m, $\theta = 20°$), and 24 m at the Abbot ice shelf site ($\Delta h = 21$ m, $\theta = 5°$). These values correspond to 0.5, 5, and 1.6 pixels in the ASTER platform, respectively. In a typical application of the shadow-height method for building height estimation ($\Delta h = 40$ m, $\theta = 50°$), this value goes down to 0.6 m; much lower, but still around the pixel size of high-resolution imagery.

Other authors have considered this effect by establishing illumination threshold values determined using image histograms, through visual inspection, or by creating a calibrated template of the shadow profile (see, e.g., [2]). For ice shelf freeboard calculations at solar elevations down to 5.2°, Guan et al. [14] have used the local maximum of the second derivative of the illumination profile to identify the shadow edge. While these methods represent a step forward to improve the identification of the edge of a shadow, none guarantee that the center of the shadow is selected. Remember that the center of the shadow corresponds to the point across the edge of the shadow that is aligned with the projector and the center of the Sun. Identifying the shadow center is crucial, as the solar elevation used in all shadow-height applications refers to the elevation of the center of the Sun, which can differ by up to 16 arcminutes from its upper or lower edges. Figure 10 shows how errors in solar elevation of this magnitude can lead to significant errors in shadow-derived heights. We addressed this problem by using a physical approach to model the illumination profile across the edge of the shadow, then searching for the image section that best fits this profile.

The second-largest error source is atmospheric refraction, which is often neglected and affects the apparent position of the Sun. This effect is especially strong at low solar elevations, as shown in Figure 9. Ignoring atmospheric refraction can easily lead to errors between 1 to 10% for solar elevations in the range used in shadow-height studies. Figure 10 shows the errors in the shadow-height method if the solar elevation angles have deviations in the range typically produced by atmospheric refraction (see Figure 9).

Figures 16 and 18 show how the error in height difference determination changes if any of these two factors is neglected. While in the Andes and Sentinel Range, the effect of atmospheric refraction is small due to the relatively high solar elevations used. We can see that using the preliminary positions of shadows identified by visual inspection can significantly increase the error and introduce a very significant bias of the order of 10 m. This suggests that what appears to the eye as the center of the shadow is closer to the projector than the actual center. This effect generates a bias towards shorter shadows and highlights the importance of our fine adjustment of shadow positions by fitting a physically derived shadow profile to the image data.

In contrast, on the Abbot ice shelf, both atmospheric refraction and the fine adjustment of shadow positions have a similar and significant contribution to the accuracy of the solution. Although, atmospheric refraction becomes particularly important on the January 2002 scene, which has the lowest solar elevation (3.7°). It is important to note that the bias toward shorter shadows persists for preliminary shadow positions and that neglecting atmospheric refraction generates a bias in the same direction.

Our careful consideration of atmospheric refraction and shadow center positioning might explain why we obtained smaller errors than previous shadow-height studies on ice shelves. Guan et al. [14] have reported freeboard height errors below 1 m for 64.1% of the measurements and below 2 m for 86.9% of measurements, when using an image with a solar elevation angle of 5.2°. In our case, with a solar elevation of 5.1° (4 January 2005 scene), 84.1% of measurements had an error below 1 m, and 100% of measurements had an error below 1.3 m. Although, this higher accuracy could also be due to differences in the characteristics of the field sites.

The Solar Position Algorithm (SPA) [36] used by Guan et al. [14] does include an atmospheric refraction calculation, based the on Sæmundsson formula [20], which takes into account surface temperature and pressure; however, the SPA implementation uses annual local average values for pressure and temperature. While this approach is reasonable for the solar radiation applications in the scope of the SPA, it fails to account for the marked seasonality of Antarctic weather.

At the Abbot ice shelf validation site, a significant part of the error came from an offset affecting all measurements on each zone. In Figure 18, we can see that the measurements were scattered around a mean value different than zero and with an absolute value between 0.48 m and 1.46 m. Arguably, this offset was partially due to snow accumulation or changes in the presence or thickness of sea ice between the time of the shadow-height measurement and the ICESat data. This bias is consistent with the variations in freeboard height computed from the seven ICESat profiles available at Zone 2, which varied by up to 0.87 m. If this offset was caused exclusively by such changes, and we assume that the measured mean value is the true freeboard height, then we could say that 100% of our measurements had errors below 1.1 m (0.7 m for measurements at the January 2005 scene only). An alternative way to characterize the error is considering the standard deviation of measurements as a gauge of the typical error. At the Abbot ice shelf site, the standard deviation of measurements ranged between 0.22 and 0.72 m. The biggest standard deviation corresponded to the January 2002 scene (see Figure 18b), likely due to a reduced performance of the shadow selection procedure associated with irregular surface reflectivity. This irregularity can be clearly seen in Figure 17, where the sea surface in the January 2005 scene (Figure 17a) looks much smoother than that in the January 2002 scene (Figure 17b).

In Section 2.8, we suggested that the error of the shadow-height method takes the form of Equation (5). This error is, thus, composed of a bias which is a function of the solar elevation and a constant $\beta$, and a relative error, which is proportional to the slope $\alpha$ and the measured height difference $\Delta h$. Considering the *a priori* estimation of $\alpha$ in Table 1 ($\alpha = 0.0018$), the relative component of the error would be negligible on ice shelves, where typical freeboard heights range between 10 and 100 m ($\sim$20 m at Abbot ice shelf). Therefore, we can attribute the error observed at the Abbot ice entirely to the bias error. In this case, measured errors were consistent with a bias constant of $\beta \geq 14.57$ m, similar to our *a priori* estimation of 14.25 m; however, if we ignore the measurement offset by considering that it is the result of snow accumulation or changes in the presence or thickness of sea ice, then the resulting bias constant would be $\beta \geq 7.84$ m. We consider that these two values are reasonable upper and lower bounds for the bias constant at the Abbot ice shelf validation site; that is, 7.84 m $\leq \beta \leq$ 14.57 m.

For the Andes and Sentinel Range validation sites, we observed a relatively small offset, compared with the spread of the measurements around the mean value (see Figure 16). This large variability of errors can be partially attributed to the different geometries associated with each measurement, leading to different sensitivities to errors in the input variables. It is important to remark that all these errors are likely overestimates of the actual errors, due to the changes in snow accumulation or glacier thickness between the image acquisition date and the field measurements. All GNSS positioning errors also contribute to increasing the maximum differences between computed and reference values.

In contrast to the Abbot ice shelf site, at the Andes and Sentinel Range sites, we cannot neglect either of the two error components described in Equation (5). To find the bias constant and slope that better represent the measured errors, we can divide Equation (5) by $\tan \theta$, in order to transform it into a linear relationship. Figure 20 shows this linear relationship for all of the mountain validation points (for the Andes and Sentinel Range data sets). The best linear fit to measured errors corresponded to a bias constant $\beta = 2.7$ m and a slope of $\alpha = 0.0022$. In total, 60% of observed errors were below this best fit, and all errors were below the *a priori* error estimate outlined in Table 1, which is represented by a dotted line in Figure 20.

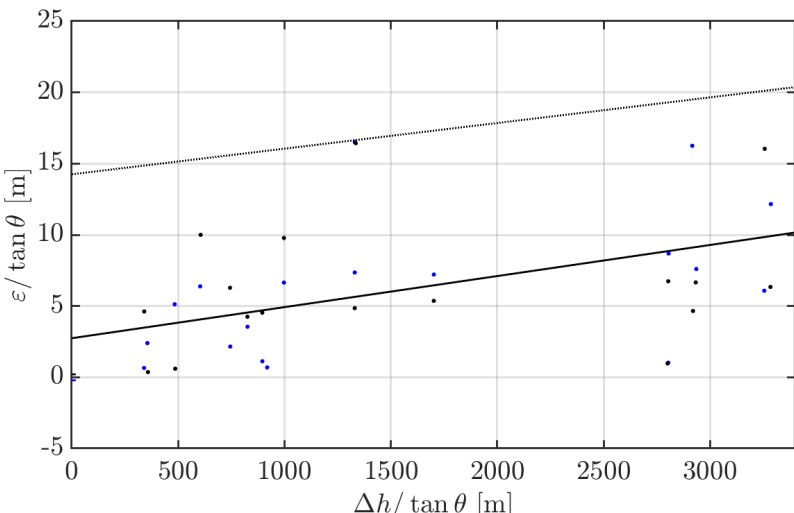

**Figure 20.** Absolute errors measured in the Andes and Sentinel range, relative to the tangent of the solar elevation $\theta$. Measurement errors represented as blue points correspond to nadir-looking bands and black points to backward-looking bands. The solid black line shows the best linear fit, and the dotted black line represents the *a priori* total error estimate detailed in Table 1.

The error slope found in Figure 20 ($\alpha = 0.0022$) was slightly larger than our *a priori* estimate in Table 1 ($\alpha = 0.0018$). This underestimation is consistent with the fact that the validation process tends to overestimate the error of the shadow-height methodology. This overestimation results from ascribing all the error sources to the methodology, not only those strictly related to it. We do not attempt to separate methodological from validation errors, as this would be difficult and somewhat arbitrary. We, thus, rather overestimate than underestimate the errors of our shadow-height methodology; however, if we stick to this larger relative error slope $\alpha = 0.0022$, we find that, for a bias constant $\beta = 9.9$ m, 94% of the measurements would fall below the resulting error estimate. This value of $\beta$ is consistent and near the middle of the range found for the Abbot ice field site. In this context, we consider that a good representation of the errors we have encountered in our validation effort is as follows:

$$\mathcal{E}_{94\%} \leq 9.9 \tan\theta [m] + 0.0022\Delta h, \tag{6}$$

where $\Delta h$ is the magnitude of the measured height difference and $\theta$ is the solar elevation. Note that, assuming the simplest geometry shown in Figure 1, a bias constant $\beta = 9.9$ m corresponds to an error in the length of the shadow of 0.66 pixels. This error is consistent and significantly smaller than our *a priori* estimate of 0.95 pixels shown in Table 1, suggesting that the combined error of the projector and shadow selection is less than expected—probably around 0.6 pixels.

In most Earth science applications, the error bias will be the largest component of the overall error. Figure 21 shows the magnitude of this component of the error as a function of the apparent solar elevation $\theta$ for three different values of the bias constant $\beta$. We can see that, for the best $\beta$ representing our validation measurements (black line), the bias error for the Andes site is between 6 and 9 m, between 2 and 5 m for the Sentinel range, and between 0.64 to 0.88 m for the Abbot ice shelf.

Figure 21, clearly show how errors in the shadow length have a lesser impact for low solar elevations. This effect might partially explain why work related to deriving building heights using the shadow-height method often report relative errors much larger than ours and other works on ice shelves, such as that of Guan et al. [14]. On the other hand, low solar elevations present fuzzier shadows, making it more challenging to accurately determine the shadow center; however, we transformed this difficulty into an advantage by modeling the illumination profile across the edge of shadows (see Section 2.5.1).

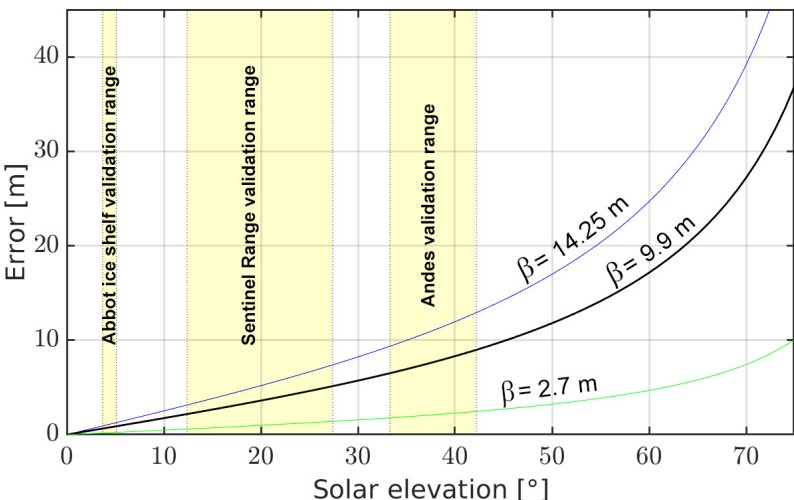

**Figure 21.** Magnitude of the bias error as a function of the apparent solar elevation $\theta$ for three different values of the bias constant $\beta$. Yellow shading shows the range of solar elevations used for each validation site.

Table 5 summarizes the main characteristics of our three validation sites, as well as the upper limit of the errors observed at each site. All errors were below ~6% of the measured height difference. We observed the largest relative error for the freeboard measurements at the Abbot ice shelf site; however, it should be noted that this site is the most sensitive to height changes between image acquisitions and reference measurements. Therefore, we estimate that the actual error was most likely around ~3–4%.

**Table 5.** Summary of upper estimates of errors observed at the three validation sites.

| Site | Latitude [°] | Solar Elevation Range [°] | Height Difference Range [m] | Error Upper Bound [m] |
|---|---|---|---|---|
| Andes | −33 | 33.3–42.2 | 308–656 | 7 ($\lesssim$2%) |
| Sentinel range | −78 | 12.4–27.4 | 292–1495 | 9 ($\lesssim$2%) |
| Abbot ice self | −72 | 3.7–5.1 | 21–22 | 1.3 ($\lesssim$6%) |

While the presented shadow-height methodology can be extended to analog imaging platforms, it is important to note that we have assumed that we can calculate the solar position with high accuracy. However, this accuracy relies on knowing the exact time at which the image was acquired. For aerial imagery and analog-era spaceborne imagery such as Corona, Hexagon, or ex-URSS declassified imagery, spacecraft location information and image acquisition time might not exist or remain classified. In these cases, the acquisition time could be inferred from the image itself, once it has been properly geolocated using ground control points. For aerial images, the time could be extracted from the shadow azimuth [7,11]. For spaceborne images, in addition to using the shadow azimuth, the timing could also be inferred by solving the orbital trajectory using a series of spacecraft locations from different image acquisitions. The spacecraft location is often obtained as part of the image geolocation process. Additionally, some analog spaceborne platforms, such as the U.S. Hexagon, include marks on the image film that encode the state of an internal timer [37], providing additional information to estimate the exact acquisition time of the image.

The overall error resulting from our validation effort (see Equation (6)) gives us confidence in the applicability of the shadow-height method to other problems, such as the measurement of the height of clouds and volcanic plumes. There are numerous images showing volcanic eruptions on ASTER, Landsat, SPOT, Sentinel-2, and other medium-resolution imaging platforms. Moreover, many more are available for lower-resolution

platforms, such as the Moderate Resolution Imaging Spectroradiometer (MODIS). This platform has a maximum ground resolution of 250 m, which would be sufficient to compute volcanic plumes heights with errors below 100 m if the solar elevation is less than 30°.

The software tools that we have developed streamline the usage of this shadow-height methodology on ASTER L1A image products. We chose the ASTER platform, as it was considered ideal for developing and validating the presented methodology. However, more work needs to be done to broaden the scope of these tools to other platforms, especially Landsat, Hexagon, and aerial photos, which would allow for the recovering of topographic information from a time, which would be especially useful for applications such as glacier mass balance.

## 5. Conclusions

In this paper, we successfully computed height differences using shadows present in satellite imagery. The height difference accuracy depends on geometry, satellite position, measurement of the shadow length, and atmospheric refraction.

The proposed shadow-height methodology uses a geometric approach that considers the movement of the satellite platform, performs accurate shadow and projector positioning using a physical model of their illumination profiles, and implements a precise refraction calculation based on archived meteorological data. The improvements in atmospheric refraction calculation and shadow positioning are novel and resulted in a substantial accuracy improvement on our validation sites.

We measured shadow-derived height differences in three validation sites, for a total of 91 independent validation measurements derived from shadows observed in ASTER L1A imagery (15 m ground resolution). We compared these results against field differential GNSS measurements or ICESat laser altimetry data. Validation data suggest that the proposed methodology has errors smaller than the horizontal resolution of the image for solar elevations below 56°, and that the accuracy improves progressively at lower solar elevations. In relative terms, all errors were below ∼6% of the measured height difference (see Table 5), with the largest values probably associated to sources related to the validation procedure, not to the shadow-height methodology.

We are confident that the reported errors are most likely overestimates of the actual errors. In general, we can estimate the expected error for an arbitrary solar elevation and height difference using Equation (6). Note that, according to this equation, absolute errors increase along with the solar elevation $\theta$ and height difference $\Delta h$, while relative errors ($\varepsilon/\Delta h$) increase with the solar elevation $\theta$ but decrease with $\Delta h$. Therefore, we observed the largest relative errors at the Abbot ice shelf, due to the small $\Delta h$ found at that site.

Higher spatial resolution satellite imagery will produce much smaller errors, as the more significant error source is proportional to the pixel size. Furthermore, the accuracy would improve with the enhanced sensor stability and orbit determination of such platforms.

We found that the accuracy of the shadow-height method is sufficient to produce useful data for many earth science problems, such as long-term glacier thickness variations, volcanic plume heights, volcanic structures change, ice shelf and iceberg freeboard height, and cloud height. This methodology can be extended to other platforms, such as Landsat 1–8, Corona, Hexagon, and aerial imagery. Extracting shadow-height measurements from such sources would contribute to the recovery of valuable terrain change information, in order to better constrain past and present surface processes.

**Supplementary Materials:** The following supporting information can be downloaded at: https://www.mdpi.com/article/10.3390/rs14071702/s1, Codes in Matlab language, GNSS measurements, ASTER L1A imagery and metadata (translated to Matlab binary format), and ICESat data are available in the Supplementary Materials.

**Funding:** This research was funded by the Comisión Nacional de Investigación Científica y Tecnológica (CONICYT) Beca de Magister 2005 grant number (folio) 22060402.

**Data Availability Statement:** The data presented in this study and Matlab codes used are available in the Supplementary Material.

**Acknowledgments:** To The Omega Foundation to support this research in Antarctica and facilitated the Trimble 5700 GNSS receiver used for the validation process and for providing Antarctic ASTER imagery. I am thankful, in particular, to my expedition partners Damien Gildea, Rodrigo Fica, Manuel Bugueño, Steve Chaplin, María Paz Ibarra, Jed Brown, and Jarmila Tyrril. To Andrés Rivera, from CECS, Chile, who facilitated ASTER imagery, provided very useful guidelines and advice, and accepted to be part of my thesis committee. To Michael Ramsey, who facilitated ASTER imagery of volcanic plumes. To my master thesis committee Emilio Vera, Rene Garreaud, and Andrés Pavez. To my family, for supporting this work in many different ways, for their advice, and motivation. And finally, to Natalia Martinez, for her support in the central Andes validation fieldwork and mainly for being the most wonderful source of light and energy in my life.

**Conflicts of Interest:** The author declares no conflict of interest.

## Abbreviations

The following abbreviations are used in this manuscript:

| | |
|---|---|
| ASTER | Advanced Spaceborne Thermal Emission and Reflection Radiometer |
| AUSPOS | Geoscience Australia GPS data processing facility |
| DEM | Digital Elevation Model |
| GDAS1 | Global Data Assimilation System one-degree |
| GIPSY | GNSS-Inferred Positioning SYstem |
| GNSS | Global Navigation Satellite System |
| LiDAR | Light Detection And Ranging |
| ICESat | Ice, Cloud, and land Elevation Satellite |
| InSAR | INterferometric Synthetic Aperture Radar |
| MODIS | MODerate resolution Imaging Spectroradiometer |
| NOAA | National Oceanic and Atmospheric Administration |
| OPUS | Online Positioning User Service |
| SCOUT | Scripps COordinate Update Tool |
| SPA | Solar Position Algorith |
| SRTM | Shuttle Radar Topography Mission |

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
