# Peer review of "High-Precision Measurement of Height Differences from Shadows in Non-Stereo Imagery: New Methodology and Accuracy Assessment"

_remotesensing, doi:10.3390/rs14071702_

Round 1
Reviewer 1 Report
Dear Authors,
I have reviewed the paper entitled "High-precision measurement of height differences from shadows in non-stereo imagery: new methodology and accuracy assessment". The paper presents "an enhanced shadow-height methodology offering significant accuracy improvement". The paper is well orgnized and written. I have only some minor remarks.
- In my opinion the Author should be careful in first person writing.
- I think the Authos could add more references for better understanding.
- Some Figures (f.e. Fig. 3) are blurry a little.
Author Response
Dear Reviewer,
Thanks for your review and feedback.
Regarding your suggestions, I've checked the first-person writing and I haven't been able to identify the points that raised your concern. If you can give me more specifics or examples I would be happy to make the corresponding corrections to the manuscript.
I've added a few more references. However, I'm not sure if they improved the understanding of the points raising your concern. If the issue is not yet solved please list the points where more references are recommended.
I've changed the format of some of the figures to lossless format (PNG) to improve sharpness. In the case of Figure 3, I've enlarged it, increased the font size and improved the text and layout.
Please look at the new version of the manuscript to see these changes implemented. A versión where I've highlighted all areas with changes has also been included for your convenience.
Reviewer 2 Report
This study presents an enhanced shadow-height methodology which applied a physical approach to model the illumination gradient through the edge of shadows and by leveraging meteorological data to precisely estimate atmospheric refraction. The method of this technology has been clearly stated in the context and it is reasonable. To demonstrate its effectiveness, this study has validated the shadow derived height at 91 site using various source of data. The new method is promising in many earth science problems and can be easily transferred to other satellite platforms.
In this article, the objectives, method, results, and conclusions have been clearly stated. However, I suggest to add a contrast experiment with traditional shadow derived height method, to quantitively show the advancement of this new method. I recommend a small revision and this article is worthy of being accepted for publication in Remote Sensing.
Author Response
Dear Reviewer,
Thanks for your review and feedback.
Regarding your suggestions, I've included two contrast experiments. The first neglects atmospheric refraction and the second doesn't use the proposed method to find the position of the center of the shadow accurately. For this last experiment, I used the preliminary positions of the shadows briefly mentioned in the first version of the manuscript as part of the methodology. These preliminary positions are found by visual inspection and represent the "traditional shadow derived height method", although there are many variants of the method that use alternative approaches.
The results of these experiments were included in figures 16 and 18, and were referred to in the Results, Discussion and Conclusions. Also, some additional details were included in the Methods to clarify the terminology and procedures that differentiate the preliminary positions of the shadows from the final ones improved with the proposed methodology.
Please look at the new version of the manuscript to see these changes implemented. A versión where I've highlighted all areas with changes has also been included for your convenience.
Reviewer 3 Report
The present work provides a methodological improvement over previous implementations by presenting a careful treatment of atmospheric refraction and a physical approach to shadow modeling and fitting. The first methodological contribution is related to the treatment of atmospheric refraction and leveraging re-analysis atmospheric data to constrain its magnitude. The second most significant methodological contribution is a physical approach to identify the exact locations of shadows in images. This approach takes advantage of the diffuse nature of shadow edges to achieve fine sub-pixel positioning accuracy. A set of experiments have been employed to validate the effectiveness of proposed method and its possible error sources.
Generally speaking, this manuscript is well motivated and written. The results could be valuable for studying surface elevation changes in present and old imagery and extending glacier variation time-series. I would like to recommend to accept it for publication under minor revision.
Minor Points: Page 2 line 33-34: ” The shadow-height method takes advantage of the basic geometric properties of the shadows often present in satellite and aerial imagery.”
Author Response
Dear Reviewer,
Thanks for your review and feedback.
Regarding your suggestions, I have to admit that I'm not sure if I've identified the error in the sentence you mention. I've changed it to "The shadow-height method takes advantage of the basic geometrical properties of shadows often present in satellite and aerial imagery." I hope this change solved the issue that raised your concern. Otherwise, please provide some specifics and I would be happy to make the corresponding corrections.
I will mention that following the suggestion of another reviewer and the editor, I've included two contrast experiments. The first neglects atmospheric refraction and the second doesn't use the proposed method to find the position of the center of the shadow accurately. I hope those help to highlight accuracy improvements achieved by the proposed methodology.
Please look at the new version of the manuscript to see these changes implemented. A versión where I've highlighted all areas with changes has also been included for your convenience.